





# EcoGEnIE 0.1: Plankton Ecology in the cGENIE Earth system model

Ben A. Ward[1,2], Jamie D. Wilson[1], Ros M. Death[1], Fanny M. Monteiro[1],
Andrew Yool[2], and Andy Ridgwell[1,3]

[1]School of Geographical Sciences, University of Bristol, Bristol BS8 1SS, UK
[2]National Oceanography Centre, European Way, Southampton SO14 3ZH, UK
[3]Department of Earth Sciences, University of California, Riverside CA, USA

**Abstract.** We present an extension to the cGENIE Earth System model that explicitly accounts for the growth and interaction of an arbitrary number of plankton species. The new package ('ECOGEM') replaces the implicit, flux-based, parameterisation of the plankton community currently employed, with explicitly resolved plankton populations and ecological dynamics. In ECOGEM, any number

of plankton species, with ecophysiological traits (e.g. growth and grazing rates) assigned according to organism size and functional group (e.g. phytoplankton and zooplankton) can be incorporated at run-time. We illustrate the capability of the marine ecology enabled Earth system model ('EcoGENIE') by comparing results from one configuration of ECOGEM (with eight generic phytoplankton and zooplankton size classes) to climatological and seasonal observations. We find that the new eco-

logical components of the model show reasonable agreement with both global-scale climatological and local-scale seasonal data. We also compare EcoGENIE results to a the existing biogeochemical incarnation of cGENIE. We find that the resulting global-scale distributions of phosphate, iron, dissolved inorganic carbon, alkalinity and oxygen are similar for both iterations of the model. A slight deterioration in some fields in EcoGENIE (relative to the data) is observed, although we make

no attempt to re-tune the overall marine cycling of carbon and nutrients here. The increased capabilities of EcoGENIE in this regard will enable future exploration of the ecological community on much longer timescales than have previously been examined in global ocean ecosystem models and particularly for past climates and global biogeochemical cycles.

## 1 Introduction

The marine ecosystem is an integral component of the Earth system and its dynamics. Photosynthetic plankton ultimately support almost all life in the ocean, including the fisheries that provide essential nutrition to more than half the human population (Hollowed et al., 2013). In addition, the marine biota determine an important downward flux of carbon, known as the 'biological pump'. This flux arises as biomass generated by photosynthesis in the well-lit ocean surface sinks into the dark ocean

interior, where it is remineralised (Hülse et al., 2017). Modulated by the activity and composition of





marine ecosystems, the biological pump increases the partial pressure of $CO_2$ at depth and decreases it in the ocean surface and atmosphere, and thus plays a key role in the regulation of Earth's climate. For instance, the existence of the biological carbon pump has been estimated to be responsible for an approximately 200 ppm decrease in atmospheric carbon concentration at steady state (Parekh et al.,

2006), with variations in its magnitude being cited as playing a key role in, for example, the late Quaternary glacial-interglacial climate oscillations (Watson et al., 2000; Hain et al., 2014).

A variety of different marine biogeochemical modelling approaches have been developed in an attempt to understand how the marine carbon cycle functions and its dynamical interaction with climate, and to make both past and future projections. In the simplest of these approaches, the bio-

logical pump is incorporated into an ocean circulation (or box) model without explicitly including any state-variables for the biota. Such models have been described as models of 'biogenically induced chemical fluxes' (rather than explicitly of the biology - and ecology - itself; Maier-Reimer, 1993). They vary considerably in complexity, but can be broadly divided into two categories. In the first of these – 'nutrient-restoring' – models calculate the biological uptake of nutrients at any one

point at the ocean surface as the flux required to maintain surface nutrient concentrations at observed values (e.g. Bacastow and Maier-Reimer, 1990; Najjar et al., 1992). The vertical flux is then remineralised at depth according to some attenuating profile, such as that of Martin et al. (1987). Within this framework, carbon export is typically calculated from the nutrient flux according to a fixed stoichiometric ('Redfield') ratio (Redfield, 1934). In addition to the availability of a spatially explicit (in

the case of ocean circulation models) observed surface ocean nutrient field, nutrient restoring models inherently only require a single parameter – the restoring time-scale, and even this parameter is not critical (as long as the time-scale is sufficiently short that the model closely reproduces the observed nutrient concentrations). The simplicity of this approach lends itself to being able to focus on a very specific part of the ecosystem dynamics, namely the downward transport of organic matter, and was

highly influential particularly during the early days of marine biogeochemical model development and assessment of carbon uptake and transport dynamics (e.g. Marchal et al., 1998; Najjar et al., 1992). However, because this approach is based explicitly upon observed values (or modified observations), they are primarily only suitable for diagnostic and modern steady-state applications and are unable to model any deviations of nutrient cycling, and hence of climate, from the current ocean

state.

More sophisticated models of biogenically induced chemical fluxes do away with a direct observational constraint and instead estimate the organic matter export term on the basis of limiting factors, such as temperature, light and the availability of nutrients such as nitrogen, phosphorous and iron – an approach we will here refer to as 'nutrient-limitation'. Models based on this approach

(e.g. Bacastow and Maier-Reimer, 1990; Heinze et al., 1991; Archer and Johnson, 2000) were natural successors to the early nutrient restoring models and could account for the influence of multiple limiting nutrients and even implicitly partition export between different functional types (Watson





et al., 2000). Without entraining an explicit dependence on observed surface ocean nutrient distributions, these models also now gain much more freedom and with it, a degree of predictive capability.

Additionally, other than plausible values for nutrient half-saturation constants, nutrient-limitation models make few assumptions that are specifically tied to modern observations, and assume very little (if anything) about the particular organisms present. Hence, as long as one makes the assumption that the marine plankton that existed at some specific time in the past were physiologically similar, particularly in terms of fundamental nutrient requirements, there is no apparent reason why nutrient-

limitation models will not be as applicable to much of the Phanerozoic in terms of geological past, as they are to the present (questions of how suitable they might be to the present in the first place, aside). Using nutrient-limitation flux schemes, marine biogeochemical cycles have hence already been simulated for periods such as the mid Cretaceous (Monteiro et al., 2012) and end Permian (Meyer et al., 2008), times for which surface nutrient distributions are not known *a priori*.

The disadvantage of both variants of models of biogenically induced chemical fluxes, is that they are not able to represent interactions between parts of the ecosystem (e.g. resource competition and predator-prey interactions), simply because these components and processes are not resolved. Nor can they address questions involving the addition or loss, such as associated with past extinction events, of plankton species and changes in ecosystem complexity and/or structure. They also suffer

from being overly responsive to changes in nutrient availability. In the case of restoring models this is simply because any change in the target field will be closely tracked. In the case of the nutrient-limitation models, the lack of an explicit biomass term results in export fluxes changing instantaneously in response to changing limiting factors. In the real world, by contrast, sufficient biomass must first exist, such as in a bloom condition, in order to achieve maximal export. This has

consequences for the how the seasonality of organic matter export is represented. Other restrictions include the inability to know anything about ecosystem size structure (and, by association, about particle sinking speed), or the degree of recycling at the ocean surface and hence the partitioning of carbon into dissolved vs. particulate phases in exported organic matter.

To allow models to respond to changes in ecosystem structure, and to incorporate some of the ad-

ditional feedbacks and complexities that may be important in determining the future marine response to continued greenhouse gas emissions (Le Quéré et al., 2005), it has been necessary to explicitly resolve the ecosystem itself. Such models have been developed across a wide range of complexities (Kwiatkowski et al., 2014). Among the simplest are 'NPZD' type models, resolving a single nutrient, homogenous phytoplankton and zooplankton communities, and a single detrital pool (Wroblewski

et al., 1988; Oschlies, 2001). At the other end of the spectrum, more complex models may include multiple nutrients and several 'Plankton Functional Types' (PFTs) (e.g. Aumont et al., 2003; Moore et al., 2002; Le Quéré et al., 2005). What links these models is that the living state variables are very broadly based on ecological guilds (i.e. groups of organisms that exploit similar resources).





While simple NPZD models are capable of reproducing some of the observed variability in bulk
properties such as chlorophyll biomass and primary production (Schartau and Oschlies, 2003b; Yool
et al., 2013; Ward et al., 2013), their very simplicity precludes the representation of many potentially
important biogeochemical processes and climate feedbacks. Additionally, NPZD models are param-
eterised to represent the activity of diverse plankton communities, with different parameter values
being required as the ecosystem changes in space and time (Schartau and Oschlies, 2003a; Losa
et al., 2006). In this regard, PFT models may be more generally applicable because they resolve rel-
atively more fundamental ecological processes that may be less sensitive to environmental variability
(Friedrichs et al., 2007). These are the key factors that have motivated the development of more com-
plex models, in which the broad ecological guilds of NPZD models are replaced with more specific
groups based on ecological and/or biogeochemical function (Aumont et al., 2015; Butenschön et al.,
2016). It is argued that resolving more components of the ecosystem allows the representation of
important climate feedbacks that cannot be accounted for in simpler models (Le Quéré, 2006).

However, alongside their advantages, the current generation of PFT models are faced with two im-
portant and conflicting challenges. Firstly, these complex models contain a large number of param-
eters that are often poorly constrained by observations (Anderson, 2005). Secondly, although PFT
models resolve more ecological structure than the preceding generation of ocean ecosystem mod-
els, they are rarely general enough to perform well across large environmental gradients (Friedrichs
et al., 2006, 2007; Ward et al., 2010). To these, one might add difficulties in their application to past
climates. PFT models are based on a conceptual reduction of the modern marine ecosystem to its
apparent key biogeochemical components, such as nitrogen fixation, or opal frustule production (as
by diatoms). The role of diatoms and the attendant cycling of silica quickly becomes moot once one
looks back in Earth history as the origin of diatoms is thought to be sometime early in the Meso-
zoic (252-66 Ma) and they did not proliferate and diversify until later in the Cenozoic (66-0 Ma)
(Falkowski et al., 2004). In addition, the physiological details of each species encoded in the model
are taken directly from laboratory culture experiments of isolated strains (Le Quéré et al., 2005)
creating a parameter-dependence on modern cultured species, in addition to a structural one.

Recent studies have begun to address these issues by focussing on the more general rules that
govern diversity (rather than by trying to quantify and parameterise the diversity itself). These 'trait-
based' models are beginning to be applied in the field of marine biogeochemical modelling (e.g.
Follows et al., 2007; Bruggeman and Kooijman, 2007), with a major advantage being that they are
able to resolve greater diversity with fewer specified parameters. One of the main challenges of this
approach then is to identify the general rules or trade-offs that govern competition between organ-
isms (Follows et al., 2007; Litchman et al., 2007). These trade-offs are often strongly constrained
by organism size. A potentially large number of different plankton size classes can therefore be pa-
rameterised according to well known allometric relationships linking plankton physiological traits
to organism size (e.g. Tang, 1995; Hansen et al., 1997). This approach has the associated advan-





tage that the size composition of the plankton community affects the biogeochemical function of the community (e.g. Guidi et al., 2009). If one assumes that the same allometric relationships and trade-offs are relatively invariant with time, then this approach provides a potential way forward to addressing geological questions.

In this paper we present an adaptable modelling framework with an ecological structure that can be easily adapted according to the scientific question at hand. The model is formulated so that all plankton are described by the same set of equations, and any differences are simply a matter of parameterisation. Within this framework, each plankton population is characterised in terms its size-dependent traits and its distinct functional type. The model also includes a realistic physiological

component, based on a cell quota model (Caperon, 1968; Droop, 1968) and a dynamic photoacclimation model (Geider et al., 1998). This physiological component increases model realism by allowing phytoplankton to flexibly take up nutrients according to availability, rather than according to an unrealistically rigid cellular stoichiometry. Such flexible stoichiometry is rarely included in large-scale ocean models, and provides the opportunity to study the links between plankton physi-

ology, ecological competition, and biogeochemistry. This model is then embedded within an Earth system model (cGENIE) widely used in addressing questions of past climate and carbon cycling, and the overall properties of the model system are evaluated.

The structure of this paper is as follows. In Section 2 we will briefly outline the nature and properties of the cGENIE Earth system model, focussing on the ocean circulation and marine biogeochem-

ical modules most directly relevant to the simulation of marine ecology. In Section 3, we introduce the new ecological model – ECOGEM – that has been developed within the cGENIE framework. Section 4 describes the preliminary experiments of ECOGEM, and Section 5 presents results from the new integrated global model (EcoGEnIE) in comparison to observations (where available) as well as to the pre-existing biogeochemical simulation of cGENIE.

## 2    The GENIE/cGENIE Earth system model

GENIE is an 'Earth system model of intermediate complexity' (EMIC) (Claussen et al., 2002) and is based on a modularised framework that allows different components of the Earth system, including ocean circulation, ocean biogeochemistry, deep-sea sediments and geochemistry, to be incorporated (Lenton et al., 2007). The simplified atmosphere and carbon-centric version of GENIE we use –

cGENIE – has been previously applied to explore and understand the interactions between biological productivity, biogeochemistry and climate over a range of timescales and time periods (e.g., Ridgwell and Schmidt, 2010; Monteiro et al., 2012; Norris et al., 2013; John et al., 2014; Gibbs et al., 2015; Meyer et al., 2016; Tagliabue et al., 2016). As is common for EMICs, cGENIE features a decreased spatial and temporal resolution in order to facilitate the efficient simulation of the various

interacting components. This imposes limits on the resolution of ecosystem dynamics to large-scale



annual/seasonal patterns in contrast to higher resolutions often used to model modern ecosystems. However, our motivation for incorporating a new marine ecosystem module into cGENIE is to focus on the explicit interactions between ecosystems, biogeochemistry and climate that are computationally prohibitive in higher resolution models. In other words, our motivation is to include and explore a more complete range of interactions and dynamics within the marine system, at the expense of spatial fidelity and with a greater intention to explore long timescale and paleoceanographic questions, rather than short-term and future anthropogenic concerns.

### 2.1 Ocean physics and climate model component – C-GOLDSTEIN

The fast climate model, C-GOLDSTEIN features a reduced physics (frictional geostrophic) 3-D ocean circulation model coupled to a 2-D energy-moisture balance model of the atmosphere and a dynamic-thermodynamic sea-ice model. Full descriptions of the model can be found in Edwards and Marsh (2005a) and Marsh et al. (2011).

The circulation model calculates the horizontal and vertical transport of heat, salinity, and biogeochemical tracers via the combined parameterisation for isoneutral diffusion and eddy-induced advection (Edwards and Marsh, 2005a; Marsh et al., 2011). The ocean model is configured on a $36 \times 36$ equal-area horizontal grid with 16 logarithmically spaced z-coordinate levels. The horizontal grid is generally constructed to be uniform in longitude ($10°$ resolution) and uniform in the sine of latitude (varying in latitude from $\sim 3.2°$ at the equator to $19.2°$ near the poles). The thickness of the vertical grid increases with depth, from 80.8 m at the surface, to as much as 765 m at depth. The degree of spatial and temporal abstraction in C-GOLDSTEIN results in parameter values that are not well known and require calibration against observations. The parameters for C-GOLDSTEIN were calibrated against annual mean climatological observations of temperature, salinity, surface air temperature and humidity using the ensemble Kalman filer (EnKF) methdology (Hargreaves et al., 2004; Ridgwell et al., 2007a). The parameter values for C-GOLDSTEIN used are those reported for the 16-level model in Table S1 of Cao et al. (2009) under "GENIE16". C-GOLDSTEIN is run with 96 time-steps per year. The resulting circulation is dynamically similar to that of classical GCMs based on the primitive equations but is significantly faster to run and in this configuration performs well against standard tests of circulation models such as anthropogenic $CO_2$ and CFC uptake, as well as reproducing the deep ocean radiocarbon ($\Delta^{14}C$) distribution (Cao et al., 2009).

### 2.2 Ocean biogeochemical model component – BIOGEM

Transformations and spatial redistribution of biogeochemical compounds both at the ocean surface (by biological uptake) and in the ocean interior (remineralisation), plus air-sea gas exchange, are handled by the module BIOGEM. In the pre-existing version of BIOGEM the biological (soft-tissue) pump is driven by an implicit (i.e. unresolved) biological community (in place of an explicit representation of living microbial community). It is therefore a nutrient limitation variant of a model of

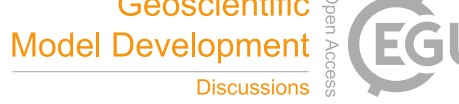



biogenically induced chemical fluxes, as outlined above. A full description can be found in (Ridgwell et al., 2007a; Ridgwell and Death, in prep.).

In this study, we use a seasonally insolation forced, 16-level ocean model configuration, similar to that of Cao et al. (2009). However, in the particular biogeochemical configuration we use, limitation

of biological uptake of carbon is provided by the availability of two nutrients. In addition to phosphate, we now include an iron cycle following (Tagliabue et al., 2016). This aspect of the model is determined by a revised set of parameters controlling the iron cycle (Ridgwell and Death, in prep.). We also incorporate a series of minor modifications to the climate model component, particularly in terms of the ocean grid and wind velocity and stress forcings (consistent with Marsh et al., 2011)

together with associated changes to several of the physics parameters. A complete description and evaluation of the physical and biogeochemical configuration of cGENIE is provided in (Ridgwell and Death, in prep.).

## 3 Ecological model component – ECOGEM

The current BIOGEM module in cGENIE does not explicitly resolve the biological community and

instead transforms surface inorganic nutrients directly into export:

- inorganic nutrients $\xrightarrow[\text{and export}]{\text{production}}$ DOM and remineralised nutrients

This simplification greatly facilitates the efficient modelling of the carbon cycle over long time scales, but with the associated caveats of an implicit scheme (as discussed earlier). In ECOGEM, biological uptake is again limited by light, temperature and nutrient availability, but here it must

pass through an explicit and dynamic intermediary plankton community, before being returned to DOM or dissolved inorganic nutrients:

- inorganic nutrients $\xrightarrow{\text{production}}$ living biomass $\xrightarrow{\text{export}}$ DOM and remineralised nutrients

The ecological community is also subject to respiration, mortality and internal trophic interactions, and will produce both inorganic compounds and organic matter. The structural relationship between

BIOGEM and ECOGEM is illustrated in Figure 1.

In the following section we outline the key state variables directly relating to ecosystem function (Section 3.1), describe the mathematical form of the key rate processes relating to each state variable (Section 3.2) and how they link together (Section 3.3). We will then describe the parameterisation of the model according to organism size and functional type (Section 3.4). The model equations

are modified from Ward et al. (2012). We provide all the equations used in ECOGEM here, but we provide only brief descriptions of the parameterisations and parameter value justifications already included in Ward et al. (2012).



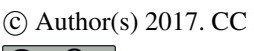

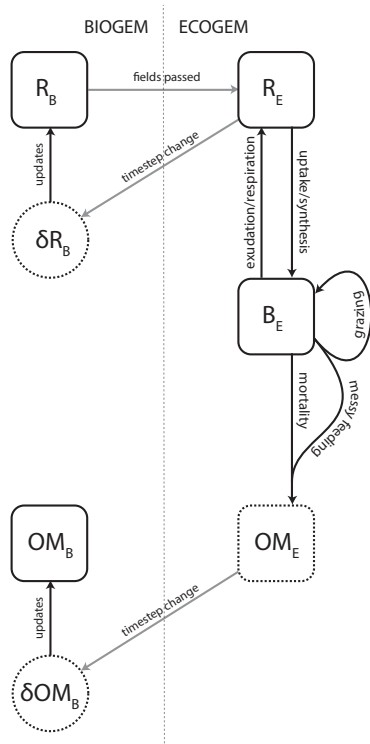

**Figure 1.** Schematic representation of the coupling between BIOGEM and ECOGEM. State variables: R = inorganic element (i.e. resource), B = plankton biomass, OM = organic matter. Subscripts $_B$ and $_E$ denote state variables in BIOGEM and ECOGEM, respectively. BIOGEM passes resource biomass R to ECOGEM. ECOGEM passes rates of change ($\delta$) in R and OM back to BIOGEM.

### 3.1 State variables

ECOGEM state variables are organised into three matrices (Table 1), representing ecologically-

relevant biogeochemical tracers (hereafter referred to as 'nutrient resources'), plankton biomass and organic matter. All these matrices have units of mmol element m$^{-3}$, with the exception of the dynamic chlorophyll quota, which is expressed in units of mg chlorophyll m$^{-3}$. The nutrient resource matrix ($\mathbf{R}$) includes $I_r$ distinct inorganic resources. The plankton community ($\mathbf{B}$) is made up of $J$ individual populations, each associated with $I_b$ cellular nutrient quotas. Finally organic matter ($\mathbf{D}$)

is made up of $K$ size classes of organic matter, each containing $i_d$ organic nutrient element pools. (Note that strictly speaking, detrital organic matter is not explicitly resolved as a state variable in ECOGEM as we currently only resolve the production of organic matter, which is passed to BIO-GEM and held there as a state variable. As a consequence, there is no grazing on detrital organic




matter in the current configuration of EcoGENIE. We include a description of **D** and its relationships
here for completeness and for convenience of notation.)

**Table 1.** State variable index notation.

| State variable | Dimensions | Index | Size | Available elements |
|---|---|---|---|---|
| **R** | Resource element | $i_r$ | $I_r$ | $DIC,\quad PO_4,\quad Fe$ |
| **B** | Plankton class | $j$ | $J$ | $1,\quad 2,\quad \dots,\quad \mathbf{J}$ |
| | Cellular quota | $i_b$ | $I_b$ | $C,\quad P,\quad Fe,\quad Chl$ |
| **D** | Organic matter size class | $k$ | $K$ | $DOM,\quad POM$ |
| | Detrital nutrient element | $i_d$ | $I_d$ | $C,\quad P,\quad Fe$ |

### 3.1.1 Inorganic resources

**R** is a row vector of length $I_r$, the number of dissolved inorganic nutrient resources.

$$\mathbf{R} = \left[ \begin{array}{ccc} DIC & PO_4 & Fe \end{array} \right] \tag{1}$$

An individual inorganic resource is denoted by the appropriate subscript. For example, $PO_4$ is de-
noted $R_{PO_4}$.

### 3.1.2 Plankton biomass

**B** is a $J \times I_b$ matrix, where $J$ is the number of plankton populations and $I_b$ is the number of cellular
quotas, including chlorophyll.

$$\mathbf{B} = \left[ \begin{array}{cccc} B_{1,C} & B_{1,P} & B_{1,Fe} & B_{1,Chl} \\ B_{2,C} & B_{2,P} & B_{2,Fe} & B_{2,Chl} \\ \vdots & \vdots & \vdots & \vdots \\ B_{J,C} & B_{J,P} & B_{J,Fe} & B_{J,Chl} \end{array} \right] \tag{2}$$

Each population and element is denoted by an appropriate subscript. For example, the total carbon
biomass of plankton population $j$ is denoted $B_{j,C}$, while the chlorophyll biomass of that population
is denoted $B_{j,Chl}$. The column vector describing the the carbon content of all plankton populations
is denoted $\mathbf{B}_C$.

   This framework can account for competition between (in theory) any number of different plankton
populations. The model equations (below) are written in terms of an 'ideal' planktonic form, with
the potential to exhibit the full range of ecophysiological traits (among those that are included in
the model). Individual populations may take on a realistic subset of these traits, according to their
assigned 'plankton functional type' (PFT) (see Section 3.4.1). Each population is also assigned a
characteristic size, in terms of equivalent spherical diameter (ESD) or cell volume. Organism size
plays a key role in determining each population's ecophysiological traits (see Section 3.4.2).





### 3.1.3 Organic detritus

**D** is a $K \times I_d$ matrix, where $K$ is the number of detrital size classes and $I_d$ is the number of detrital nutrient elements.

$$\mathbf{D} = \begin{bmatrix} D_{1,C} & D_{1,P} & D_{1,Fe} \\ D_{2,C} & D_{2,P} & D_{2,Fe} \end{bmatrix} \tag{3}$$

Each size class and element is denoted by an appropriate subscript. For example, dissolved organic phosphorus (size class $k=1$) is denoted $D_{1,P}$, while particulate organic iron (size class $k=2$) is denoted $D_{2,Fe}$.

### 3.2 Plankton physiology and ecology

The rates of change in each state variable within ECOGEM are defined by a range of ecophysiolog-

ical processes. These are defined by a set of mathematical functions that are common to all plankton populations. Parameter values are defined in Section 3.4.

#### 3.2.1 Temperature limitation

Temperature affects a wide range of metabolic processes through an Arrehnius-like equation that is here set equal for all plankton.

$$\gamma_{\mathrm{T}} = e^{A(\mathrm{T}-T_{\mathrm{ref}})} \tag{4}$$

The parameter $A$ describes the temperature sensitivity, T is the ambient water temperature in degrees C, and $T_{\mathrm{ref}}$ is a reference temperature (also in degrees C) at which $\gamma_{\mathrm{T}} = 1$.

#### 3.2.2 The plankton 'quota'

The physiological status of a plankton population is defined in terms of its cellular nutrient quota, **Q**,

which is the ratio of assimilated nutrient (phosphorus or iron) to carbon biomass. For each plankton population, $j$, and each planktonic quota, $i_b (\neq C)$,

$$Q_{j,i_b} = \frac{B_{j,i_b}}{B_{j,C}} \tag{5}$$

This equation is also used to describe the population chlorophyll content relative to carbon biomass. The size of the quota increases with nutrient uptake, chlorophyll synthesis, or the loss of carbon. The

quota decreases through the acquisition of carbon (described below).

Excessive accumulation of P or Fe biomass in relation to carbon is prevented as the uptake or assimilation of each nutrient element is down-regulated as the respective quota becomes full. The generic form of the uptake regulation term for element $i_b$ is given by a linear function of the nutrient status, modified by an additional shape-parameter ($h$=0.1) that allows greater assimilation under





low-to-moderate resource limitation.

$$Q_{j,i_b}^{\text{stat}} = \left( \frac{Q_{j,i_b}^{\max} - Q_{j,i_b}}{Q_{j,i_b}^{\max} - Q_{j,i_b}^{\min}} \right)^h \tag{6}$$

### 3.2.3 Nutrient uptake

Phosphate and dissolved iron ($i_r = i_b = $ P or Fe) are taken up as functions of environmental availability ($[R_{i_r}]$), maximum uptake rate ($V_{j,i_r}^{\max}$), the nutrient affinity ($\alpha_{j,i_r}$), the quota satiation term, ($Q_{j,i_b}^{\text{stat}}$) and temperature limitation ($\gamma_{\text{T}}$):

$$V_{j,i_r} = \frac{V_{j,i_r}^{\max} \alpha_{j,i_r} [R_{i_r}]}{V_{j,i_r}^{\max} + \alpha_{j,i_r} [R_{i_r}]} Q_{j,i_b}^{\text{stat}} \cdot \gamma_{\text{T}} \tag{7}$$

This equation is effectively equivalent to the Michaelis-Menten type response, but replaces the half-saturation constant with the more mechanistic nutrient affinity, $\alpha_{j,i_r}$.

### 3.2.4 Photosynthesis

The photosynthesis model is modified from Geider et al. (1998) and Moore et al. (2002). Light-limitation is calculated as a Poisson function of local irradiance ($I$), modified by the iron-dependent initial slope of the P-I curve ($\alpha \cdot \gamma_{j,\text{Fe}}$) and the chlorophyll-$a$-to-carbon ratio ($Q_{j,\text{Chl}}$).

$$\gamma_{j,I} = \left[ 1 - \exp \left( \frac{-\alpha \cdot \gamma_{j,\text{Fe}} \cdot Q_{j,\text{Chl}} \cdot I}{P_{j,\text{C}}^{\text{sat}}} \right) \right] \tag{8}$$

Here $P_{j,\text{C}}^{\text{sat}}$ is maximum light-saturated growth rate, modified from an absolute maximum rate of $P_{j,\text{C}}^{max}$, according to the current nutrient and temperature limitation terms.

$$P_{j,\text{C}}^{sat} = P_{j,\text{C}}^{max} \cdot \gamma_T \cdot \min \left[ \gamma_{j,\text{P}}, \ \gamma_{j,\text{Fe}} \right] \tag{9}$$

The nutrient-limitation term is given as a minimum function of the internal nutrient status (Droop, 1968; Caperon, 1968; Flynn, 2008), each defined by normalised hyperbolic functions for P and Fe ($i_b = $ P or Fe),

$$\gamma_{j,i_b} = \frac{1 - Q_{j,i_b}^{\min}/Q_{j,i_b}}{1 - Q_{j,i_b}^{\min}/Q_{j,i_b}^{\max}}, \tag{10}$$

The gross photosynthetic rate ($P_{j,\text{C}}$) is then modified from $P_{j,\text{C}}^{sat}$ by the light-limitation term.

$$P_{j,\text{C}} = \gamma_{j,I} P_{j,\text{C}}^{\text{sat}} \tag{11}$$

Net carbon uptake is given by

$$V_{j,\text{C}} = P_{j,\text{C}} - \xi \cdot V_{j,\text{P}} \tag{12}$$

With the second term accounting for the metabolic cost of biosynthesis ($\xi$). This parameter was originally defined as a loss of carbon as a fraction of nitrogen uptake (Geider et al., 1998). We define it here relative to phosphate uptake, using a fixed N:P ratio of 16.

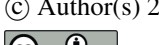



### 3.2.5 Photoacclimation

The chlorophyll-to-carbon ratio is regulated as the cell attempts to balance the rate of light capture by chlorophyll with the maximum potential (i.e. light-replete) rate of carbon fixation. Depending on this ratio, a certain fraction of newly assimilated phosphorus is diverted to the synthesis of new chlorophyll $a$,

$$\rho_{j,\mathrm{Chl}} = \theta_{\mathrm{P}}^{\max} \frac{P_{j,\mathrm{C}}}{\alpha \cdot \gamma_{j,\mathrm{Fe}} \cdot Q_{j,\mathrm{Chl}} \cdot I} \tag{13}$$

Here $\rho_{j,\mathrm{Chl}}$ is the amount of chlorophyll $a$ that is synthesised for every mmol of phosphorus assim-
ilated (mg Chl (mmol P)$^{-1}$) with $\theta_{\mathrm{P}}^{\max}$ representing the maximum ratio (again converting from the nitrogen based units of Geider et al., 1998, with a fixed N:P ratio of 16). If phosphorus is assimilated at carbon specific rate $V_{j,\mathrm{P}}$ (mmol P (mmol C)$^{-1}$ d$^{-1}$), then the carbon specific rate of chlorophyll $a$ synthesis (mg chl (mmol C)$^{-1}$ d$^{-1}$) is

$$V_{j,\mathrm{Chl}} = \rho_{j,\mathrm{Chl}} \cdot V_{j,\mathrm{P}} \tag{14}$$

### 3.2.6 Light attenuation

ECOGEM uses a slightly more complex light attenuation scheme than BIOGEM, which simply calculates a mean solar (shortwave) irradiance averaged over the depth of the surface layer, and assuming a length scale of 20 m over which light decays (Doney et al., 2006). BIOGEM then takes this mean irradiance and applies a Michaelis-Menten like limitation term, assuming a half saturation
value for light of 20 W m$^{-2}$ (Doney et al., 2006). At the ocean surface, the incoming shortwave solar radiation intensity is taken from the climate component in cGENIE and varies seasonally (Edwards and Marsh, 2005b; Marsh et al., 2011).

In ECOGEM the light level is calculated as the mean level of photosynthetically available radia-
tion within a variable mixed layer (with depth calculated according to Kraus and Turner, 1967). We
also take into account inhibition of light penetration due to the presence of light absorbing particles and dissolved molecules (Shigsesada and Okubo, 1981). If $Chl_{tot}$ is the total chlorophyll concentra-
tion in the surface layer (of thickness $Z_1$), and $Z_{ML}$ is the mixed-layer depth, the virtual chlorophyll concentration distributed across the mixed layer is given by

$$Chl_{ML} = Chl_{tot} \frac{Z_1}{Z_{ML}} \tag{15}$$

The combined light-attenuation coefficient attributable to both water and the virtual chlorophyll concentration is given by

$$k_{tot} = k_w + k_{chl} \cdot Chl_{ML} \tag{16}$$

For a given level of photosynthetically available radiation at the ocean surface ($I_0$), plankton in the surface grid box experience the average irradiance within the mixed layer, which is given by

$$I = \frac{I_0}{k_{tot}} \frac{1}{Z_{ML}} (1 - e^{(-k_{tot} \cdot Z_{ML})}) \tag{17}$$





### 3.2.7 Predation (including both herbivorous and carnivorous interactions)

Here we define predation simply as the consumption of any living organism, regardless of the trophic level the organism (i.e. phytoplankton or zooplankton prey).

The predator-biomass-specific grazing rate of predator ($j_{\mathrm{pred}}$) on prey ($j_{\mathrm{prey}}$) is given by,

$$G_{j_{\mathrm{pred}},j_{\mathrm{prey}},\mathrm{C}} = \gamma_{\mathrm{T}} \cdot \underbrace{G^{\max}_{j_{\mathrm{pred}},\mathrm{C}} \cdot \frac{\mathcal{F}_{j_{\mathrm{pred}},\mathrm{C}}}{k_{j_{prey},\mathrm{C}} + \mathcal{F}_{j_{\mathrm{pred}},\mathrm{C}}}}_{\text{overall grazing rate}} \cdot \underbrace{\Phi_{j_{\mathrm{pred}},j_{\mathrm{prey}}}}_{\text{switching}} \cdot \underbrace{\left(1 - e^{\Lambda \cdot \mathcal{F}_{j_{\mathrm{pred}},\mathrm{C}}}\right)}_{\text{prey refuge}} \tag{18}$$

where $\gamma_{\mathrm{T}}$ is the temperature-dependence, $G^{\max}_{j_{\mathrm{pred}},\mathrm{C}}$ is the maximum grazing rate, and $k_{j_{prey},\mathrm{C}}$ is the half-saturation concentration for all (available) prey. The overall grazing rate is a function of total food available to the predator, $\mathcal{F}_{j_{\mathrm{pred}},\mathrm{C}}$. This is given by the product of the prey biomass vector, $\mathbf{B}_{\mathrm{C}}$, and the grazing kernel ($\phi$),

$$\underset{[J_{\mathrm{pred}} \times 1]}{\boldsymbol{\mathcal{F}}_{\mathrm{C}}} = \underset{[J_{\mathrm{pred}} \times J_{\mathrm{prey}}]}{\boldsymbol{\phi}} \underset{[J_{\mathrm{prey}} \times 1]}{\boldsymbol{B}_{\mathrm{C}}} \tag{19}$$

Note that this equation is written out in matrix form, with the dimensions noted underneath each matrix. Each element of the grazing matrix $\phi$ is an approximately log-normal function of the predator-to-prey length ratio, $\vartheta_{j_{\mathrm{pred}},j_{\mathrm{prey}}}$, with an optimum ratio of $\vartheta_{\mathrm{opt}}$ and a geometric standard deviation $\sigma_{j_{\mathrm{pred}}}$.

$$\phi_{j_{\mathrm{pred}},j_{\mathrm{prey}}} = \exp\left[-\left(\ln\left(\frac{\vartheta_{j_{\mathrm{pred}},j_{\mathrm{prey}}}}{\vartheta_{\mathrm{opt}}}\right)\right)^2 / \left(2\sigma^2_{j_{\mathrm{pred}}}\right)\right] \tag{20}$$

We also include an optional 'prey-switching' term, such that predators may preferentially attack those prey that are relatively more available (i.e. active switching, $s = 2$). Alternatively they may attack prey in direct proportion to their availability (i.e. passive switching, $s = 1$). In the simulations below we assume active switching.

$$\Phi_{j_{\mathrm{pred}},j_{\mathrm{prey}}} = \frac{(\phi_{j_{\mathrm{pred}},j_{\mathrm{prey}}} \mathrm{B}_{j_{\mathrm{prey}},\mathrm{C}})^s}{\sum^J_{j_{\mathrm{prey}}=1}(\phi_{j_{\mathrm{pred}},j_{\mathrm{prey}}} \mathrm{B}_{j_{\mathrm{prey}},\mathrm{C}})^s} \tag{21}$$

Finally, a prey refuge function is incorporated, such that the overall grazing rate is decreased when the availability of all prey ($\mathcal{F}_{j_{\mathrm{pred}},\mathrm{C}}$) is low. The size of the prey refuge is dictated by the coefficient $\Lambda$. The overall grazing response is calculated on the basis of prey carbon. Grazing losses of other prey elements are simply calculated from their stoichiometric ratio to prey carbon, with different elements assimilated according to the predator's nutritional requirements (see below).

$$G_{j_{\mathrm{pred}},j_{\mathrm{prey}},\mathrm{i}_b} = G_{j_{\mathrm{pred}},j_{\mathrm{prey}},\mathrm{C}} \frac{\mathrm{B}_{j_{\mathrm{prey}},i_b}}{\mathrm{B}_{j_{\mathrm{prey}},\mathrm{C}}} \tag{22}$$

### 3.2.8 Prey assimilation

Prey biomass is assimilated into predator biomass with an efficiency of $\lambda_{j_{\mathrm{pred}},i_b}$ ($i_b \neq \mathrm{Chl}$). This has a maximum value of $\lambda^{\mathrm{max}}$ that is modified according the the quota status of the predator. For



elements $i_b$ = P or Fe, prey biomass is assimilated as a function of the respective predator quota. If the quota is full, the element is not assimilated. If the quota is empty, the element is assimilated with maximum efficiency ($\lambda^{\mathrm{max}}$).

$$\lambda_{j_{\mathrm{pred}},i_b} = \lambda^{\mathrm{max}} Q^{\mathrm{stat}}_{j,i_b} \tag{23}$$

C assimilation is regulated according to the status of the most limiting nutrient element (P or Fe) modified by the same shape-parameter, $h$, that was applied in Equation 6.

$$Q^{\mathrm{lim}}_{j,i_b} = \left( \frac{Q_{j,i_b} - Q^{\mathrm{min}}_{j,i_b}}{Q^{\mathrm{max}}_{j,i_b} - Q^{\mathrm{min}}_{j,i_b}} \right)^h \tag{24}$$

If both nutrient quotas are full, C is assimilated at the maximum rate. If either are empty, C assimilation is down-regulated until sufficient quantities of the limiting element(s) are acquired.

$$\lambda_{j_{\mathrm{pred}},\mathrm{C}} = \lambda^{\mathrm{max}} \min\left( Q^{\mathrm{lim}}_{j,\mathrm{P}}, Q^{\mathrm{lim}}_{j,\mathrm{Fe}} \right) \tag{25}$$

### 3.2.9 Respiration

A linear respiration rate is applied to degrade plankton carbon biomass into dissolved inorganic carbon. This is achieved through a $J$ by $I_r$ respiration matrix, $\mathbf{r}$, which is non-zero only for $i_r = \mathrm{DIC}$.

### 3.2.10 Death

All living biomass is subject to a linear mortality rate of $m_p$. This rate is decreased at very low biomasses (population carbon biomass $\lesssim 1 \times 10-6$ mmol C m$^{-3}$) in order to maintain a viable population within every surface grid cell ("everything is everywhere, but the environment selects", Baas-Becking, 1934).

$$m_j = m_p(1 - e^{-10^{10} \cdot B_{j,C}}) \tag{26}$$

The low biomass at which a population attains 'immortality' is sufficiently small for that population to have a negligible impact on all other components of the ecosystem.

### 3.2.11 Calcium carbonate

The production and export of calcium carbonate (CaCO$_3$) by calcifying plankton in the surface ocean is scaled to the export of particulate organic carbon via a spatially-uniform value which is
modified by a thermodynamically-based relationship with the calcite saturation state. The dissolution of CaCO$_3$ below the surface is treated in a similar way to that of particulate organic matter (equation 32), as described by Ridgwell et al. (2007a) with the parameter values controlling the export ratio between CaCO$_3$ and POC taken from Ridgwell et al. (2007b).



### 3.2.12 Oxygen

Oxygen production is coupled to photosynthetic carbon fixation via a fixed linear ratio, such that

$$V_{j,O_2} = -\frac{106}{138} V_{j,\text{DIC}} B_{j,\text{C}} \tag{27}$$

The negative sign indicates that oxygen is produced as DIC is consumed. Oxygen consumption associated with the remineralisation of organic matter is unchanged relative to BIOGEM.

### 3.2.13 Alkalinity

Production of alkalinity is coupled to planktonic uptake of $PO_4$ via a fixed linear ratio, such that

$$V_{j,Alk} = -16 V_{j,\text{PO}_4} \cdot B_{j,\text{C}} \tag{28}$$

The negative sign indicates that alkalinity increases as $PO_4$ is consumed. This relationship accounts for alkalinity changes associated with N transformations (Zeebe and Wolf-Gladrow, 2001) that are not explicitly represented in the biogeochemical configurations of cGENIE that are applied here.

### 430 3.2.14 Production of organic matter

Plankton mortality and grazing are the only two sources of organic matter, with partitioning between non-sinking dissolved and sinking particulate phases determined by the parameter $\beta_j$. In this initial implementation of ECOGEM, for traceability, the assumptions are the same as made in the current version of BIOGEM (Ridgwell and Death, in prep.) which themselves follow the OCMIP2 ocean carbon cycle modelling intercomparison protocol described in Najjar et al. (2007). Specifically, $\beta_j$

is set to a fixed fraction $\beta$ for all size classes.

### 3.3 Differential equations

Differential equations for $\mathbf{R}$, $\mathbf{B}$ and $\mathbf{D}$ are written below. The dimensions of each matrix and vector used equations 29 to 31 are given in Table 1. Note that while $\mathbf{R}$ and $\mathbf{OM}$ are transported by the

physical component of GENIE, living biomass $\mathbf{B}$ is not currently subject to any physical transport. The only communication between biological communities in adjacent grid cells is through the advection and diffusion of inorganic resources and non-living organic matter in BIOGEM. Note that some additional sources and sinks of $\mathbf{R}$, and all sinks of $\mathbf{D}$, are computed in BIOGEM.

### 3.3.1 Inorganic resources

For each inorganic resource, $i_r$,

$$\frac{\partial R_{i_r}}{\partial t} = \sum_{j=1}^{J} \Big( \underbrace{-V_{j,i_r} \cdot B_{j,\text{C}}}_{\text{uptake}} + \underbrace{r_{j,i_r} \cdot B_{j,\text{C}}}_{\text{respiration}} \Big) \tag{29}$$





### 3.3.2 Plankton biomass

For each plankton class, $j$, and internal biomass quota, $i_b$,

$$\frac{\partial B_{j,i_b}}{\partial t} = + \underbrace{V_{j,i_b} \cdot B_{j,C}}_{\text{uptake}} - \underbrace{m_j \cdot B_{j,i_b}}_{\text{basal mortality}} - \underbrace{r_{i_b,j} \cdot B_{C,j}}_{\text{respiration}}$$

$$+ \underbrace{B_{j,C} \cdot \lambda_{j,i_b} \sum_{j_{\text{prey}}=1}^{J} G_{j,j_{\text{prey}},i_b}}_{\text{grazing gains}} - \underbrace{\sum_{j_{\text{pred}}=1}^{J} B_{j_{\text{pred}},C} \cdot G_{j_{\text{pred}},j,i_b}}_{\text{grazing losses}} \qquad (30)$$

### 3.3.3 Dissolved organic matter

For each detrital nutrient element, $i_d$, the rate of change of dissolved fraction of organic matter ($k = 1$) is described by

$$\frac{\partial \mathrm{D}_{1,i_d}}{\partial t} = \underbrace{\sum_{j=1}^{J} [\mathrm{B}_{j,i_d}] \beta_j m_j}_{\text{mortality}} + \underbrace{\sum_{j_{\text{pred}}=1}^{J} [\mathrm{B}_{j_{\text{pred}},C}] (1-\lambda_{j_{\text{pred}},i_b}) \sum_{j_{\text{prey}}=1}^{J} \beta_{j_{\text{prey}}} G_{j_{\text{pred}},j_{\text{prey}},i_d}}_{\text{messy feeding}} \qquad (31)$$

Dissolved organic matter ($\mathbf{D}_1$) is an explicit tracer that is transported by the ocean circulation model and is degraded back to its constituent nutrients with a fixed turnover time of $\lambda$ (= 0.5 years). Particulate organic matter (POM) is not represented as an explicit state variable in either ECOGEM or BIOGEM. Instead, its implicit production in the surface layer is given by

$$F_{\text{surface},i_d} = \underbrace{\sum_{j=1}^{J} [\mathrm{B}_{j,i_d}](1-\beta_j)m_j}_{\text{mortality}} + \underbrace{\sum_{j_{\text{pred}}=1}^{J} [\mathrm{B}_{j_{\text{pred}},C}](1-\lambda_{j_{\text{pred}},i_b}) \sum_{j_{\text{prey}}=1}^{J} (1-\beta_{j_{\text{prey}}}) G_{j_{\text{pred}},j_{\text{prey}},i_d}}_{\text{messy feeding}}$$

This surface production is redistributed throughout the water column as a depth dependent flux, $F_{z,i_d}$. To achieve this, $F_{\text{surface},i_d}$ is partitioned between a 'refractory' component ($r^{\text{POM}}$) that is predominantly remineralised close to the seafloor, and a 'labile' component ($1 - r^{\text{POM}}$) which predominantly remineralises in the upper water column. The net remineralisation at depth $z$, relative to the export depth $z_0$ is determined by characteristic length scales ($l^{\text{rPOM}}$ and $l^{\text{POM}}$ for 'refractory' and 465   'labile' POM respectively):

$$F_{z,i_d} = F_{\text{surface},i_d} \left[ (1-r^{\text{POM}}) \cdot \exp(\frac{z_0-z}{l^{\text{POM}}}) + r^{\text{POM}} \cdot \exp(\frac{z_0-z}{l^{\text{rPOM}}}) \right] \qquad (32)$$

The remineralisation length scales reflect a constant sinking speed and constant remineralisation rate. All POM reaching the seafloor is remineralised instantaneously. See Ridgwell et al. (2007a) for a fuller description and justification.

### 470   3.3.4 Coupling to BIOGEM

The calculations in BIOGEM are performed 48 times for each model year (i.e. once for every 2 time-steps taken by the ocean circulation mode). ECOGEM takes 20 time steps for each BIOGEM time-step i.e. 960 time-steps per year). At the beginning of each ECOGEM time-step loop, concentrations



of inorganic tracers and key properties of the physical environment are passed from BIOGEM. The
ecological community responds by transforming inorganic compounds into living biomass through
photosynthesis. At the end of each ECOGEM time step loop, the rates of change in $\mathbf{R}$ and $\mathbf{OM}$
are passed back to BIOGEM. $\partial \mathbf{R}/\partial t$ is used to update DIC, phosphate, iron, oxygen and alkalinity
tracers, while $\partial \mathbf{D}_1/\partial t$ is added to the dissolved organic matter pools. The rate of particulate organic
matter production, $\partial \mathbf{D}_2/\partial t$ is instantly remineralised at depth using to the standard BIOGEM export
functions described above (equation 32). $\frac{\partial \mathbf{B}}{\partial t}$ is used only to update the living biomass concentrations
within ECOGEM. The structure of the coupling is illustrated in Figure 1.

In the initial implementation of ECOGEM described and evaluated here, the explicit plankton
community is held entirely within the ECOGEM module and is not subject to physical transport
(e.g. advection and diffusion) by the ocean circulation model (although dissolved tracers such as
nutrients still are). As a first approximation, this approach appears to be acceptable, as long as the
rate of transport between the very large grid cells in cGENIE is slow in relation to the net growth
rates of the plankton community. On-line advection of ecosystem state variables will be implemented
and its consequences explored in a future version of EcoGENIE.

### 3.4 Ecophysiological parameterisation

The model community is made up of a number of different plankton populations, with each one
described according to the same set of equations, as outlined above. Differences between the pop-
ulations are specified according to individual parameterisation of the equations. In the following
sections, we describe how the members of the plankton community are specified, and how their
parameters are assigned according to the organism's size and taxonomic group.

### 3.4.1 Model structure

The plankton community in ECOGEM is designed to be highly configurable. Each population
present in the initial community is specified by a single line in an input text file, which describes
the organism size and taxonomic group.

In this configuration we include 16 plankton populations across eight different size classes. These
are divided into two PFTs, namely, "Phytoplankton" and "Zooplankton" (see Table 2). The eight
phytoplankton populations have nutrient uptake and photosynthesis traits enabled, and predation
traits disabled, whereas the opposite is true for the eight zooplankton populations. In future we
expect to bring in a wider range of trait-based functional types, including siliceous plankton (e.g.
Follows et al., 2007), calcifiers (Monteiro et al., 2016), nitrogen fixers (Monteiro et al., 2010), and
mixotrophs (Ward and Follows, 2016).





**Table 2.** Plankton functional groups and sizes in the standard run.

| $j$ | PFT | ESD ($\mu$m) | $j$ | Functional Type | ESD ($\mu$m) |
|---|---|---|---|---|---|
| 1 | Phytoplankton | 0.6 | 11 | Zooplankton | 0.6 |
| 2 | Phytoplankton | 1.9 | 12 | Zooplankton | 1.9 |
| 3 | Phytoplankton | 6.0 | 13 | Zooplankton | 6.0 |
| 4 | Phytoplankton | 19.0 | 14 | Zooplankton | 19.0 |
| 5 | Phytoplankton | 60.0 | 15 | Zooplankton | 60.0 |
| 6 | Phytoplankton | 190.0 | 16 | Zooplankton | 190.0 |
| 7 | Phytoplankton | 600.0 | 17 | Zooplankton | 600.0 |
| 8 | Phytoplankton | 1900.0 | 18 | Zooplankton | 1900.0 |

### 3.4.2 Size-dependent traits

With the exception of the maximum photosynthetic rate ($P_{\mathrm{C}}^{\max}$, see below), the size-dependent eco-physiological parameters ($p$) given in Table 3 are assigned as power-law functions of organismal volume ($V = \pi[ESD]^3/6$) according to standard equations of the form,

$$p = a\left(\frac{V}{V_0}\right)^b \tag{33}$$

Here $V_0$ is a reference value of $V_0 = 1$ $\mu$m$^3$. The value of $p$ at $V = V_0$ is given by the coefficient $a$, while the rate of change in $p$ as a function of $V$ is described by the exponent $b$.

The maximum photosynthetic rate ($P_{\mathrm{C}}^{\max}$) of very small cells (i.e. $\lesssim 5$ $\mu$m ESD) has been shown to deviate from the standard power law of equation 33 (Raven, 1994; Bec et al., 2008; Finkel et al., 2010), so we use the slightly more complex unimodal function given by Ward and Follows (2016).

$$P_{\mathrm{C}}^{\max} = \frac{p_a + \log_{10}(\frac{V}{V_0})}{p_b + p_c \log_{10}(\frac{V}{V_0}) + \log_{10}(\frac{V}{V_0})^2} \tag{34}$$

The parameters of this equation (listed in Table 3), were derived empirically from the data of Marañón et al. (2013).

### 3.4.3 Size-independent traits

A list of size-independent model parameters are listed in Table 4.

### 3.5 Parameter modifications

As far as possible, the parameter values applied in ECOGEM were kept as close as possible to previously published versions of the model (Ward and Follows, 2016). There were however a few modifications that were required to bring EcoGENIE into first order agreement with observations and the current version of cGENIE (Ridgwell and Death, in prep.). In particular, in comparison to

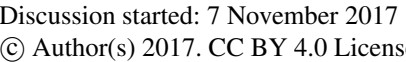



**Table 3.** Size-dependent ecophysiological parameters ($p$) and their units, with size-scaling coefficients ($a$, $b$ and $c$) for use in equations 33 and 34.

| Parameter | Symbol | Size-scaling coefficients | | | Units |
|---|---|---|---|---|---|
| | $p$ | $a$ | $b$ | $c$ | |
| Inorganic nutrient uptake | | | | | |
| Maximum photosynthetic rate | $P_{\mathrm{C}}^{\max}$ | 3.08 | 5.00 | -3.80 | mmol N (mmol C)$^{-1}$ d$^{-1}$ |
| Maximum nutrient uptake rates | $V_{\mathrm{PO_4}}^{\max}$ | $4.4\times10^{-2}$ | 0.06 | | mmol P (mmol C)$^{-1}$ d$^{-1}$ |
| | $V_{\mathrm{Fe}}^{\max}$ | $1.4\times10^{-4}$ | -0.09 | | mmol Fe (mmol C)$^{-1}$ d$^{-1}$ |
| Nutrient affinities | $\alpha_{\mathrm{PO_4}}$ | 1.10 | -0.35 | | m$^3$ (mmol C)$^{-1}$ d$^{-1}$ |
| | $\alpha_{\mathrm{Fe}}$ | 0.175 | -0.36 | | m$^3$ (mmol C)$^{-1}$ d$^{-1}$ |
| Carbon quotas | | | | | |
| Cell carbon content | $Q_{\mathrm{C}}$ | $1.45\times10^{-11}$ | 0.88 | | mmol C cell$^{-1}$ |
| Grazing | | | | | |
| Maximum prey ingestion rate | $G_{\mathrm{C}}^{max}$ | 21.9 | -0.16 | | d$^{-1}$ |

the biogeochemical model used in Ward and Follows (2016), the amount of soluble iron supplied to cGENIE by atmospheric deposition is considerably less. With a smaller source of iron, it was necessary to decrease the iron demand of the plankton community, and this was achieved by decreasing $Q_{\mathrm{Fe}}^{\max}$ and $Q_{\mathrm{Fe}}^{\min}$ by five-fold ($Q_{\mathrm{Fe}}^{\max}$ from 20 to 4 nmol Fe (mmol C)$^{-1}$, and $Q_{\mathrm{Fe}}^{\min}$ from 5 to 1 nmol Fe (mmol C)$^{-1}$).

We also found that the flexible stoichiometry of ECOGEM led to excessive export of carbon from the surface ocean, attributable to higher C:P ratios in organic matter (BIOGEM assumes a Redfieldian C:P of 106). This effect was moderated by adding the respiration term, which returns a fraction of carbon biomass directly to DIC (it is assumed that other elements are not lost in this way). The additional production of POC also led to increased production of calcium carbonate. This was counteracted by increasing the PIC:POC production ratio ($r^{\mathrm{CaCO_3:POC}}$) from 0.022 to 0.0285, and decreasing the thermodynamic calcification rate power ($\eta$) from 1.28 to 0.744 (Ridgwell et al., 2007a).

## 4 Simulations and Data

### 4.1 10,000 year spin-up

We ran cGENIE (as configured and described in Ridgwell and Death, in prep.) and EcoGENIE (as described here) each for period of 10,000 years. These runs were initialised from a homogenous and static ocean, with an imposed constant atmospheric $CO_2$ concentration of 278 ppm. We present model output from the 10,000th year of integration.





**Table 4.** Size-independent model parameters.

| Parameter | Symbol | Value | Units |
|---|---|---|---|
| **Nutrient quotas** | | | |
| Minimum phosphate:carbon quota | $Q_{\mathrm{P}}^{\min}$ | $2.1\times10^{-3}$ | mmol P (mmol C)$^{-1}$ |
| Maximum phosphate:carbon quota | $Q_{\mathrm{P}}^{\max}$ | $1.1\times10^{-2}$ | mmol P (mmol C)$^{-1}$ |
| Minimum iron:carbon quota | $Q_{\mathrm{Fe}}^{\min}$ | $1.0\times10^{-6}$ | mmol Fe (mmol C)$^{-1}$ |
| Maximum iron:carbon quota | $Q_{\mathrm{Fe}}^{\max}$ | $4.0\times10^{-6}$ | mmol Fe (mmol C)$^{-1}$ |
| **Temperature** | | | |
| Reference temperature | $T_{\mathrm{ref}}$ | 20 | °C |
| Temperature dependence | $A$ | 0.05 | - |
| **Photosynthesis** | | | |
| Maximum Chl-$a$-to-phosphorus ratio | $\theta_{\mathrm{N}}^{\max}$ | 48 | mg Chl $a$ (mmol P)$^{-1}$ |
| Initial slope of P-I curve | $\alpha$ | $3.83\times10^{-7}$ | mmol C (mg Chl $a$)$^{-1}$($\mu$Ein m$^{-2}$)$^{-1}$ |
| Cost of biosynthesis | $\xi$ | 37.28 | mmol C (mmol P)$^{-1}$ |
| **Grazing** | | | |
| Optimum predator:prey length ratio | $\vartheta_{\mathrm{opt}}$ | 10 | - |
| Geometric s.d. of $\vartheta$ | $\sigma_{\mathrm{graz}}$ | 2.0 | - |
| Total prey half-saturation | $k_{\mathrm{C}}^{\mathrm{prey}}$ | 5.0 | mmol C m$^{-3}$ |
| Maximum assimilation efficiency | $\lambda^{\max}$ | 0.7 | - |
| Grazing refuge parameter | $\Lambda$ | -1 | (mmol C m$^{-3}$)$^{-1}$ |
| Active switching parameter | $s$ | 2 | - |
| Assimilation shape parameter | $h$ | 0.1 | - |
| **Other loss terms** | | | |
| Plankton mortality | $m$ | 0.05 | d$^{-1}$ |
| Plankton respiration | $r_{i_b=\mathrm{DIC}}$ | 0.05 | d$^{-1}$ |
| | $r_{i_b\neq\mathrm{DIC}}$ | 0 | d$^{-1}$ |
| **Partitioning of organic matter** | | | |
| Fraction to DOM | $\beta$ | 0.66 | - |
| **Light attenuation** | | | |
| Light attenuation by water | $k_{\mathrm{w}}$ | 0.04 | m$^{-1}$ |
| Light attenuation by chlorophyll | $k_{\mathrm{Chl}}$ | 0.03 | m$^{-1}$(mg Chl)$^{-1}$ |



### 4.2 Observations

Although they are not necessarily strictly comparable, we compare results from the pre-industrial configurations of cGENIE and EcoGENIE to contemporary climatologies from a range of sources. Global climatologies of dissolved phosphate and oxygen are drawn from the World Ocean Atlas (WOA 2009), while DIC and alkalinity are taken from Global Ocean Data Analysis Project (GLO-DAP). Surface chlorophyll concentrations represent a climatological average from 1997 to 2002, estimated by the SeaWiFS satellite. Depth-integrated primary production is from Behrenfeld and Falkowski (1997). All of these interpolated global fields have been re-gridded onto the cGENIE $36 \times 36 \times 16$ grid.

Observed dissolved iron concentrations are those published by Tagliabue et al. (2012). These data are too sparse and variable to allow reliable mapping on the cGENIE grid, and are therefore shown as individual data.

Fidelity to the observed seasonal cycle of nutrients and biomass was evaluated against observations from nine Joint Global Ocean Flux Study (JGOFS) sites: the Hawai'i Ocean Time-series (HOT: 23°N, 158°W), the Bermuda Atlantic Time-series Study (BATS: 32°N, 64°W), the equatorial Pacific (EQPAC: 0°N, 140°W), the Arabian Sea (ARABIAN: 16°N, 62°E), the North Atlantic Bloom Experiment (NABE: 47°N, 19°W), Station P (STNP: 50°NS, 145°W), Kerfix (KERFIX: 51°S, 68°E), Antarctic Polar Frontal Zone (APFZ: 62°S, 170°W) and the Ross Sea (ROSS: 75°S, 180°W). Model output for KERFIX and the Ross Sea site was not taken at the true locations of the observations (51°S, 68°E and 75°S, 180°W, respectively). Kerfix was moved to compensate for a poor representation of the Polar Front within the coarse resolution ocean model, while the Ross Sea site does not lie within the GEnIE ocean grid. At each site, the observational data represent the mean daily value within the mixed layer. Observational data from all years are plotted together as one climatological year.

## 5 Results

### 5.1 Biogeochemical variables

We start by describing the global distributions of key biogeochemical tracers that are common to both cGENIE and EcoGENIE.

#### 5.1.1 Global surface values

Annual mean global distributions are presented for the upper 80.8 m of the water column, corresponding to the model surface layer. In Figure 2 we compare output from the two models to observations of dissolved phosphate and iron. Surface phosphate concentrations are broadly similar between the two versions of the model, except that EcoGENIE provides slightly lower estimates in the South-





ern Ocean and equatorial upwellings. Both versions strongly underestimate surface phosphate in the equatorial and north Pacific, and to a lesser extent in the north and east Atlantic, the Arctic and the

580 Arabian Sea. This is likely attributable in part to the model underestimating the strength of upwelling in these regions. It should also be noted that the observations may in some cases be unrepresentative of the true surface layer, when this is significantly shallower than 80.8 m. In such cases the observed value will be affected by measurements from below the surface layer. Iron distributions are also broadly similar between the two models, with EcoGENIE showing slightly lower iron concentra-

585 tions over most of the ocean.

Figure 3 shows observed and modelled values of inorganic carbon, oxygen and alkalinity. The two models yield very similar surface distributions of the three tracers. DIC and alkalinity are both broadly underestimated relative to observations, while oxygen shows higher fidelity, albeit with artificially high estimates in the equatorial Atlantic and Pacific. This is likely attributable to unrealis-

590 tically weak upwelling in these regions.

Surface $\Delta pCO_2$ from the two models is shown in Figure 4. EcoGENIE shows weaker $CO_2$ outgassing in tropical band, with a much stronger ocean-to-atmosphere flux in the Western Arctic.

In Figure 5 we show the annual mean rate of particulate organic matter production in the surface layer, and the relative differences between ECOGEM and BIOGEM. In comparison to cGENIE,

EcoGENIE shows elevated POC production in all regions. Production of $CaCO_3$ is globally less variable in EcoGENIE than cGENIE, with notable higher fluxes in the oligotrophic gyres and polar regions.

The relative proportions in which these elements and compounds are exported from the surface ocean are regulated by the stoichiometry of biological production. In cGENIE (BIOGEM), carbon

and phosphorus production are rigidly coupled through a fixed ratio of 106:1, while POFe:POC and $CaCO_3$:POC production ratios are regulated as a function of environmental conditions. In ecoGE-NIE (ECOGEM), phosphorus, iron and carbon production are all decoupled through the flexible quota physiology, which depends on both environmental conditions, and the status of the food-web. Only $CaCO_3$:POC production ratios are regulated via the same mechanism in the two models (al-

though we decreased the average $CaCO_3$:POC ratio in ECOGEM to compensate for the elevated POC production relative to POP).





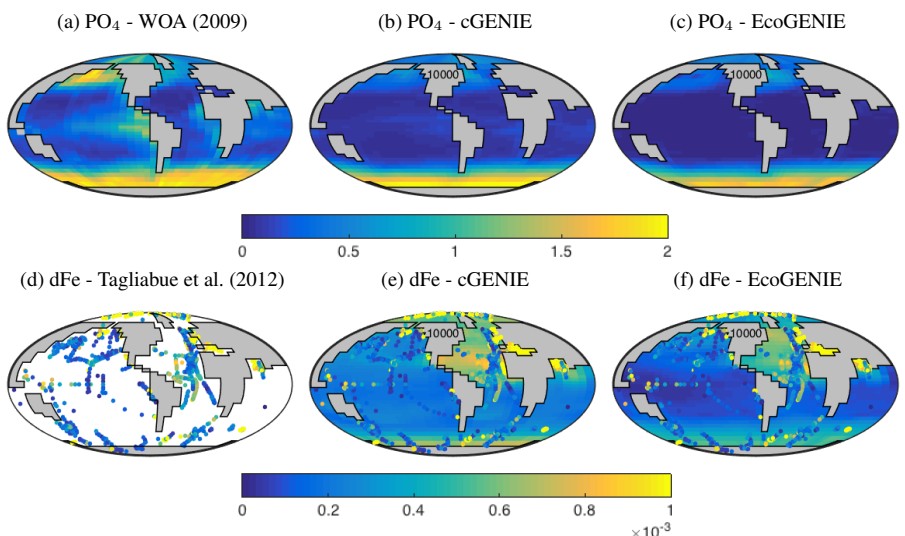

**Figure 2.** Surface concentrations of dissolved inorganic nutrients.

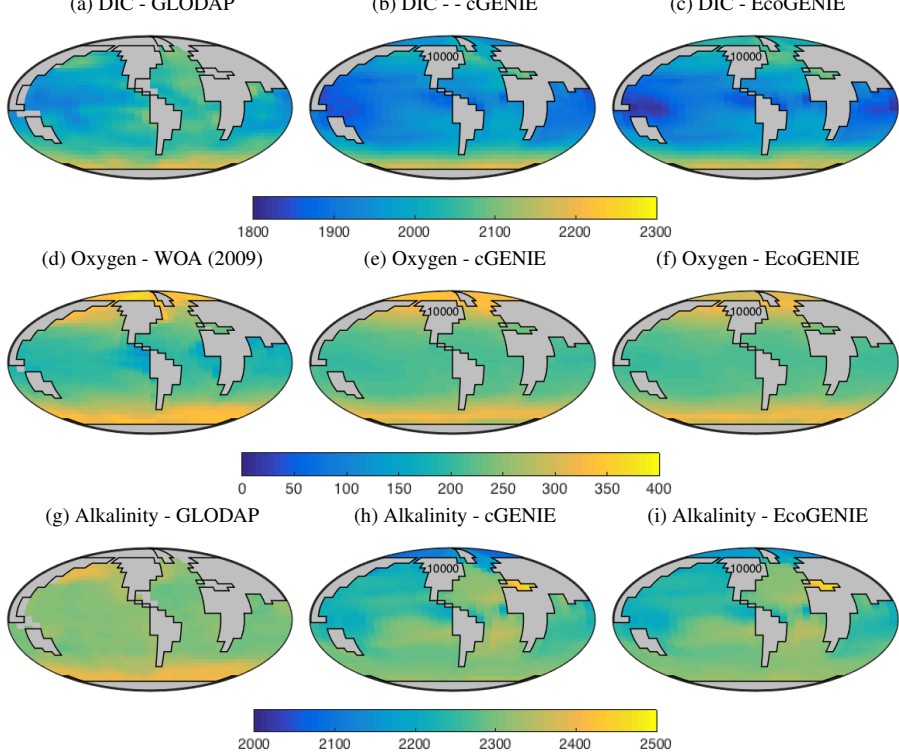

**Figure 3.** Surface concentrations of dissolved inorganic carbon, alkalinity and dissolved oxygen.





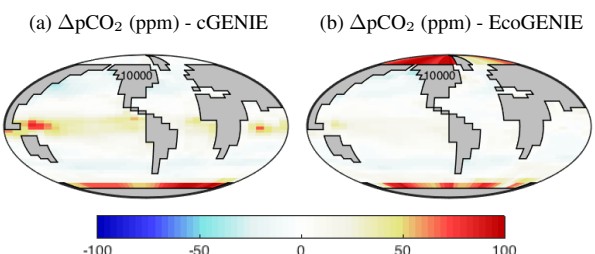

**Figure 4.** (Preindustrial) surface $\Delta pCO_2$.





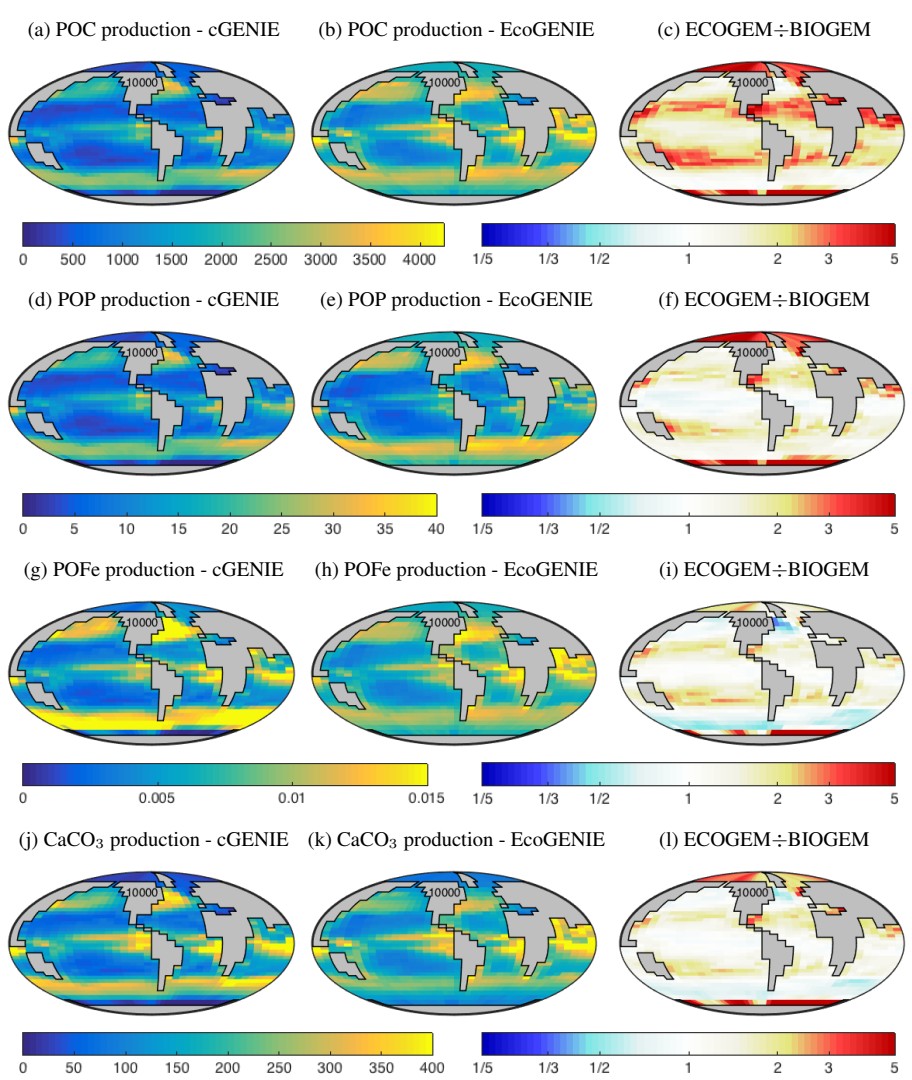

**Figure 5.** Particulate matter production (and export from the surface layer). The right-hand column indicates
the relative increase or decrease in ECOGEM, relative to BIOGEM.





### 5.1.2 Basin-averaged depth profiles

In this section we present the meridional depth distributions of key biogeochemical tracers, averaged across each of the three main ocean basins, as shown in Figure 6. Figure 7 shows that the distribution of dissolved phosphate is very similar between the two models, with EcoGENIE showing a slightly stronger sub-surface accumulation in the northern Indian Ocean.

The vertical distributions shown in Figure 8 reveal that dissolved iron is lower throughout the ocean in EcoGENIE, relative to cGENIE, particularly below 1500 m. Differences are less obvious at intermediate depths. (Observations are currently too sparse to estimate reliable basin-scale distributions of dissolved iron; see Tagliabue 2016.)

Figure 9 shows that while cGENIE reproduces observed DIC distributions very well, EcoGENIE overestimates concentrations within the Indian and Pacific Oceans. The total oceanic DIC inventory increased by just under 2% from 0.299 mol C in cGENIE to 0.304 in EcoGENIE (with a fixed atmospheric $CO_2$ concentration of 278 ppm). Otherwise the two models show broadly similar distributions, with the most pronounced differences (as for $PO_4$) in the northern Indian Ocean.

Figure 10 shows that cGENIE reasonably captures the invasion of $O_2$ into the ocean interior through the Southern Ocean and North Atlantic. These patterns are also seen in EcoGENIE, although unrealistic water column anoxia is seen in the northern intermediate Indian and Pacific Oceans. Again, this is likely a consequence of greater export and remineralisation of organic carbon in Eco-GENIE, leading to more oxygen consumption at intermediate depths (also evidenced by elevated $PO_4$, DIC and alkalinity in the same regions; Figures 7, 9 and 11).

Alkalinity (Figure 11) also shows some clear differences between the two models, again most noticeably in the northern intermediate Indian and Pacific Oceans. In these regions EcoGENIE shows excessive accumulation of alkalinity at $\sim$1000 m depth. This is again attributable to the increased C export in EcoGENIE. In the absence of a nitrogen cycle (and $NO_3^-$ reduction), increased anoxic remineralisation of organic carbon (Figures 9 and 10) leads to increased reduction of sulphate to $H_2S$, which in turn increases the alkalinity of seawater. Further adjustment of the respiration of carbon in ECOGEM and hence the effective exported P:C Redfield ratio, and/or retuning of the organic matter remineralisation profiles in BIOGEM (Ridgwell et al., 2007a) would likely resolve these issues.





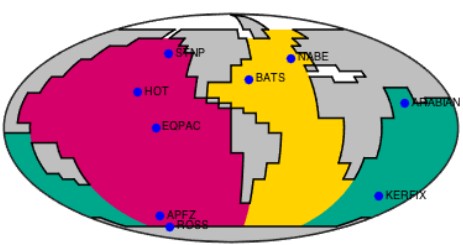

**Figure 6.** Spatial definition of the three ocean basins used in Figures 7 to 10. Locations of the JGOFS time-series sites are indicated with blue dots.

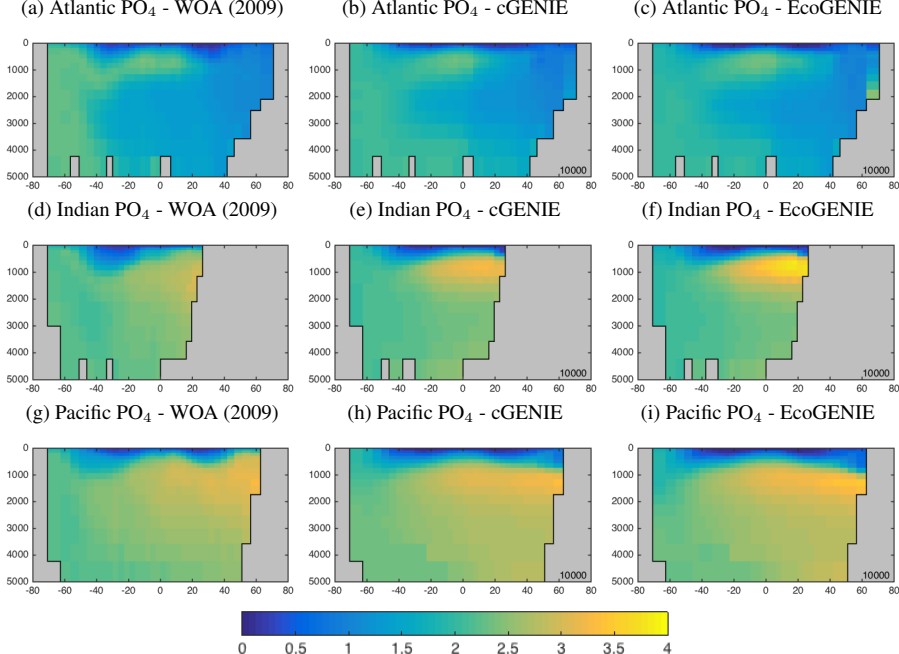

**Figure 7.** Basin-averaged meridional-depth distribution of phosphate.



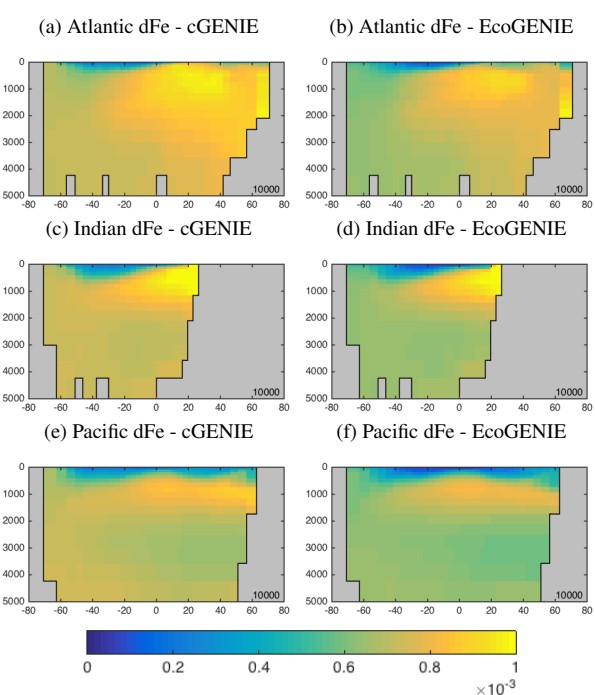

**Figure 8.** Basin-averaged meridional-depth distribution of total dissolved iron (dFe).





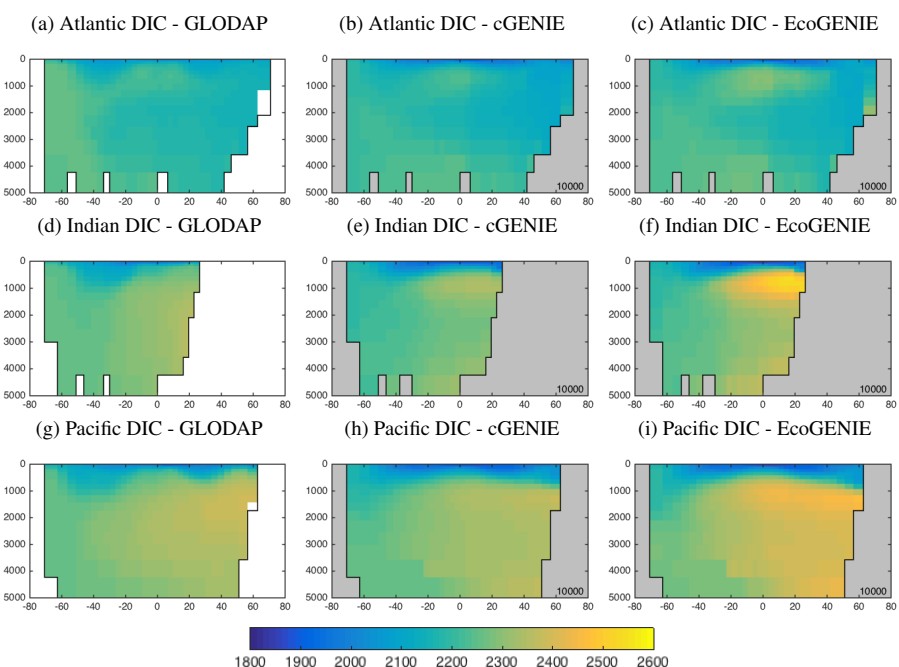

**Figure 9.** Basin-averaged meridional-depth distribution of DIC.





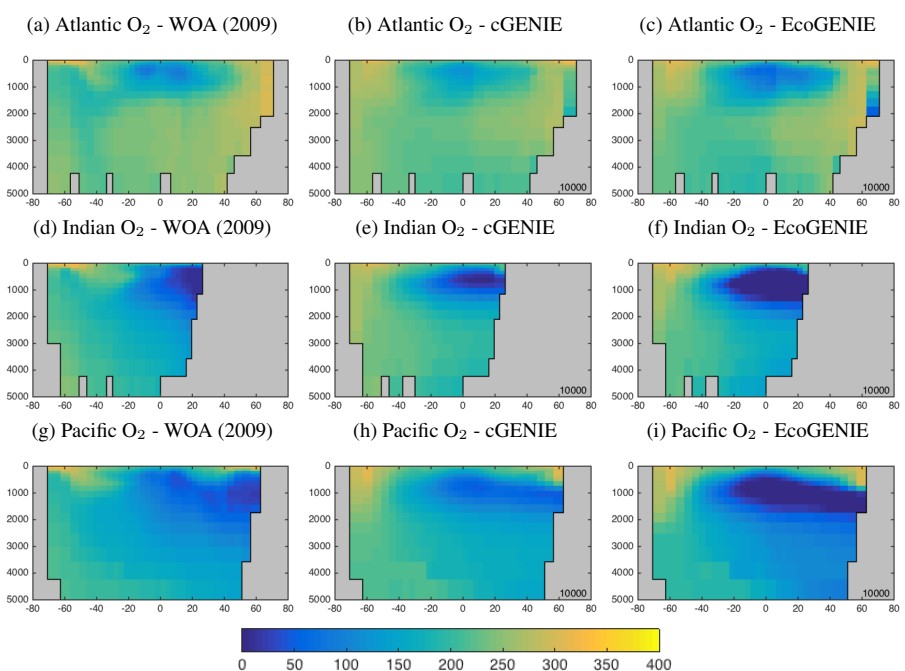

**Figure 10.** Basin-averaged meridional-depth distribution of dissolved oxygen.



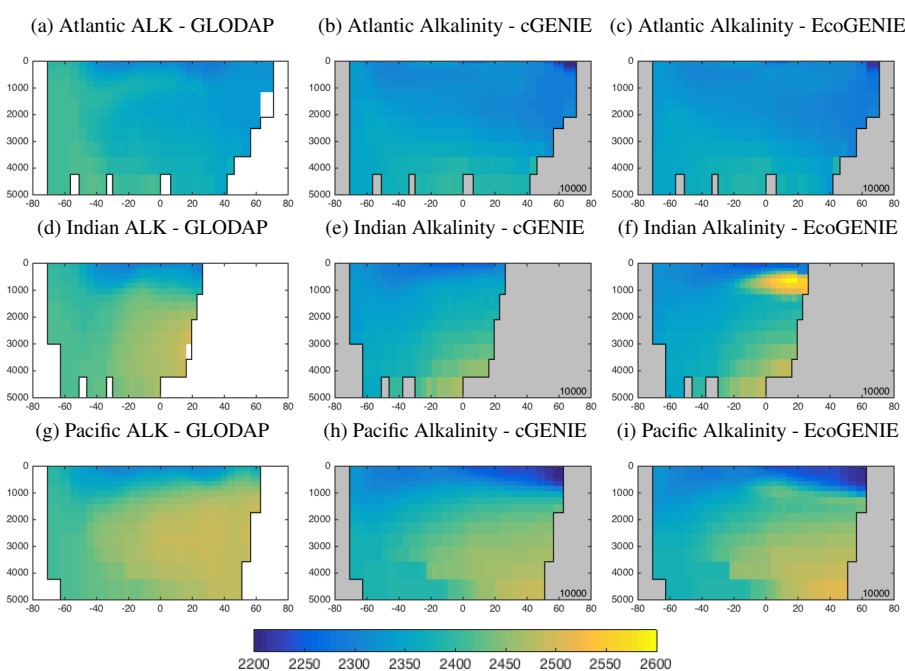

**Figure 11.** Basin-averaged meridional-depth distribution of alkalinity.





### 5.1.3 Time-series

Figures 12 and 13 we compare the seasonal cycles of surface nutrients (phosphate and iron) at nine
Joint Global Ocean Flux Study (JGOFS) sites.

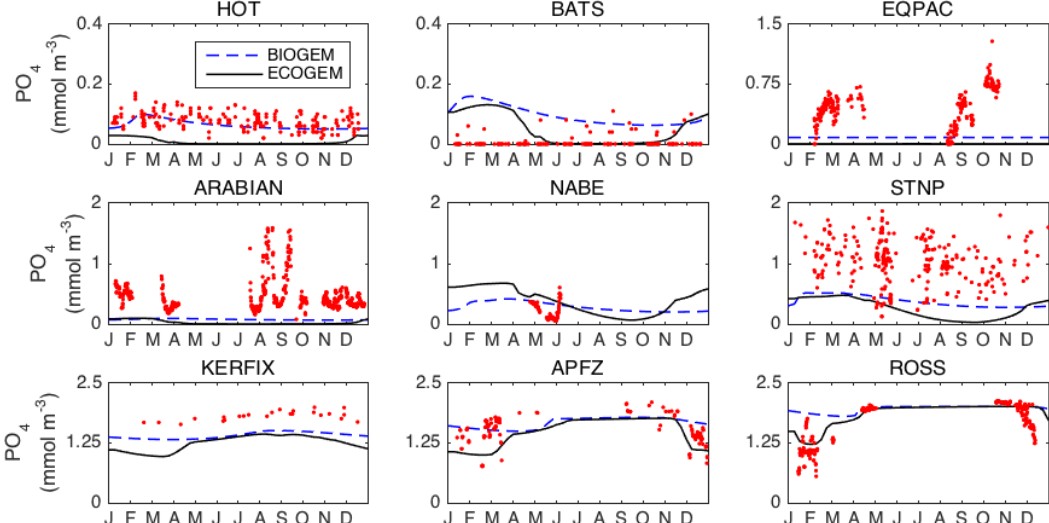

**Figure 12.** Annual cycle of surface $PO_4$ at 9 time-series sites in cGENIE and EcoGENIE. Red dots indicate
climatological observations, while the lines represent modelled surface $PO_4$ concentrations. Locations of the
time-series are indicated in Figure 6.





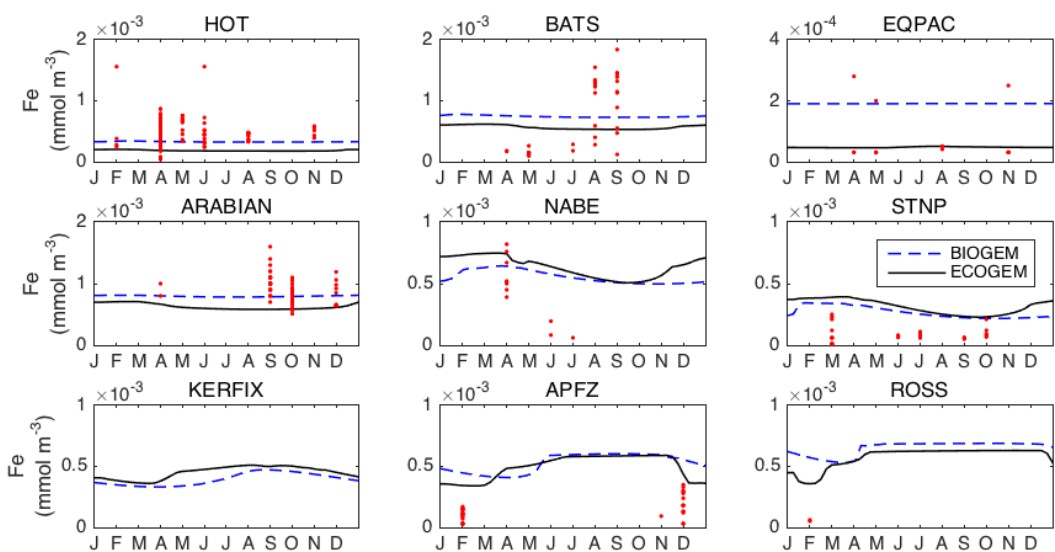

**Figure 13.** Annual cycle of surface dissolved iron at 9 time-series sites in cGENIE and EcoGENIE. Red dots indicate climatological observations, while the lines represent modelled surface iron concentrations. Locations of the time-series are indicated in Figure 6.



### 5.2 Ecological variables

Moving on from the core components that are common to both models, we present a range of ecological variables that are exclusive to EcoGENIE. As before, we begin by presenting the annual mean global distributions in the ocean surface layer, comparing total chlorophyll and primary production to satellite-derived estimates (Figure 14). We then look in more detail at the community composition, with Figure 15 showing the carbon biomass within each plankton population. Figure 16 then shows the degree of nutrient limitation within each phytoplankton population. Finally, in Figure 18, we show the seasonal cycle of community and population level chlorophyll at each of the nine JGOFS time-series sites.

### 5.2.1 Global surface values

Figure 14 reveals that EcoGENIE shows some limited agreement with the satellite-derived estimate of global chlorophyll. As expected, chlorophyll biomass is elevated in the high-latitude oceans relative to lower latitudes. The sub-tropical gyres show low biomass, but the distinction with higher latitudes is not as clear as in the satellite estimate. The model also shows a clear lack of chlorophyll in equatorial and coastal upwelling regions, relative to the satellite estimate. The model predicts higher chlorophyll concentrations in the Southern Ocean than the satellite estimate, although it should be noted that the satellite algorithms may be underestimating concentrations in these regions (Dierssen, 2010).

Modelled primary production correctly increases from the oligotrophic gyres towards high latitudes and upwelling regions, but variability is much lower than in the satellite estimate. Specifically, the model and satellite estimates yield broadly similar estimates in the oligotrophic gyres, but the model does not attain the high values seen at higher latitudes and in coastal areas.

Figure 15 shows the modelled carbon biomass concentrations in the surface layer, for each modelled plankton population. The smallest (0.6 $\mu$m) phytoplankton size class is evenly distributed in the low-latitude oceans between 40° N and S, but is largely absent nearer to the poles. The 1.9 $\mu$m phytoplankton size class is similarly ubiquitous at low latitudes, albeit with somewhat higher biomass, and its range extends much further towards the poles. With increasing size, the larger phytoplankton are increasingly restricted to highly productive areas, such as the sub-polar gyres and upwelling zones.

Perhaps as expected, zooplankton size classes tend to mirror the biogeography of their phytoplankton prey. The smallest (1.9 $\mu$m) surviving size class is found primarily at low latitudes, although a highly variable population is found at higher latitudes. This population is presumably supported by grazing on the larger 6 $\mu$m size class (with very low efficiency dictated by the unfavourable predator-prey length ratio). Larger zooplankton size classes follow a similar pattern to the phytoplankton,




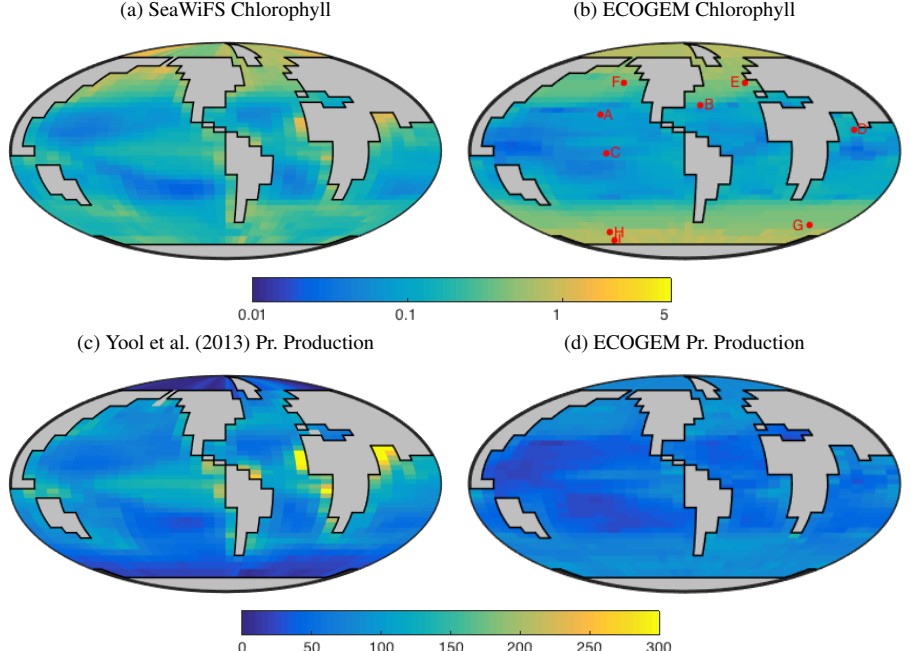

**Figure 14.** Satellite-derived (left) and modelled (right) surface chlorophyll *a* concentration and depth-integrated primary production. The satellite-derived estimate of primary production is a composite of three products (Behrenfeld and Falkowski, 1997; Carr et al., 2006; Westberry et al., 2008), as in Yool et al. (2013, their Figure 12).

moving from a cosmopolitan but homogenous distribution in the smaller size classes, towards spatially more variable distributions among the larger organisms.

The degree of nutrient limitation within each phytoplankton size class is shown in Figure 16. The two-dimensional colour-scale indicates decreasing iron limitation from left to right, and decreasing phosphorus limitation from bottom to top. White is therefore nutrient replete, blue is phosphorus limited, red is iron limited, and magenta is phosphorus-iron co-limited. The figure demonstrates that the smallest size class is not nutrient limited in any region. The increasing saturation of the colour scale in larger size classes indicates an increasing degree of nutrient limitation. As expected, nutrient limitation is strongest in the highly stratified low latitudes. A stronger vertical supply of nutrients at higher latitudes is associated with weaker nutrient limitation, although nutrient limitation is still significant among the larger size classes. Consistent with observations (Moore et al., 2013), phosphorus limitation is restricted to low latitudes. Iron limitation dominates in high latitude regions. Among the larger size classes the upwelling zones appear to be characterised by iron-phosphorus co-limitation.



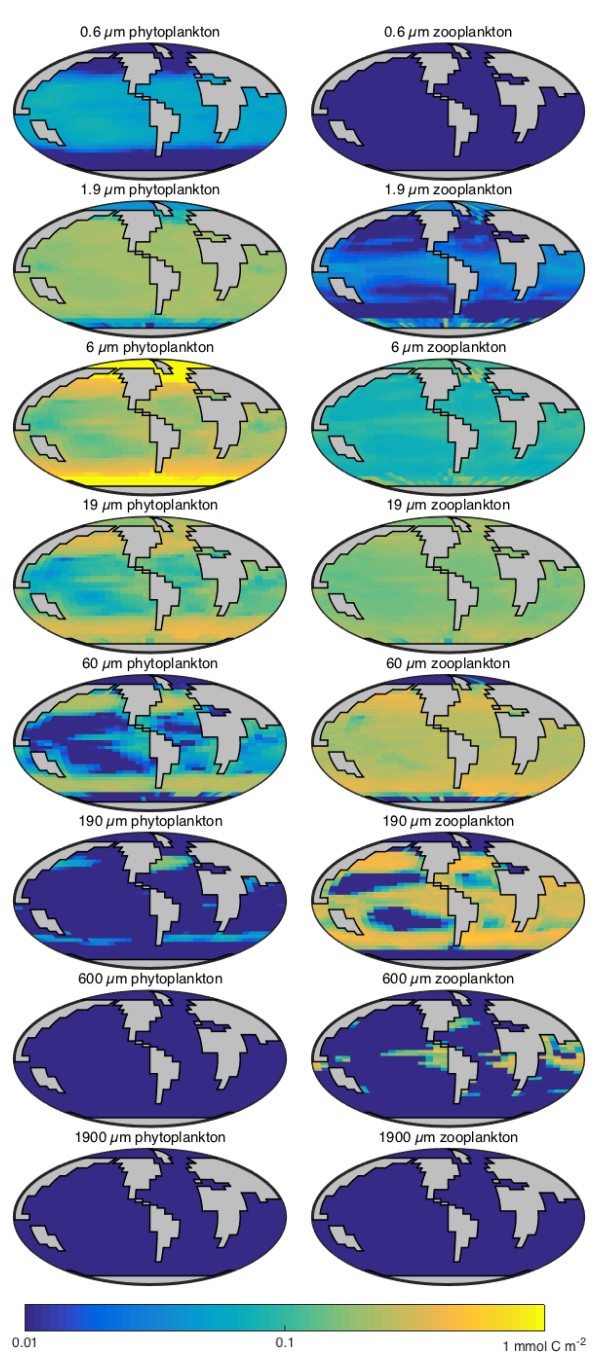

**Figure 15.** Surface concentrations of carbon biomass in each population.





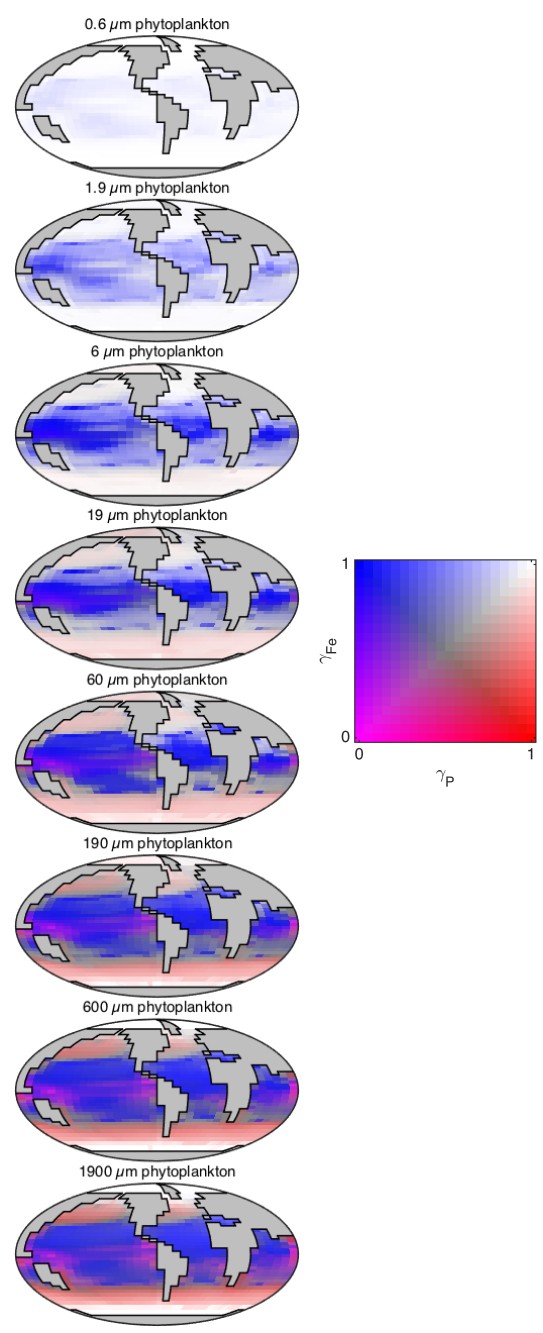

**Figure 16.** Nutrient limitation in each phytoplankton population. The two-dimensional colour-scale indicates decreasing phosphorus limitation from left to right, and decreasing iron limitation from bottom to top. White is therefore nutrient replete, blue is phosphorus limited, red is iron limited, and magenta is phosphorus-iron co-limited.



### 5.2.2 Time-series

The seasonal cycles of phytoplankton chlorophyll *a* are compared to time-series observations in Figure 18. The modelled total chlorophyll concentrations (black lines) track the observed concentrations (red dots) reasonably well at most sites, and perhaps better than might be expected from the comparison to satellite data in Figure 14. The modelled surface chlorophyll concentration is probably too low in the equatorial Pacific, while the spring bloom occurs one to two months earlier than was seen during the North Atlantic Bloom Experiment.

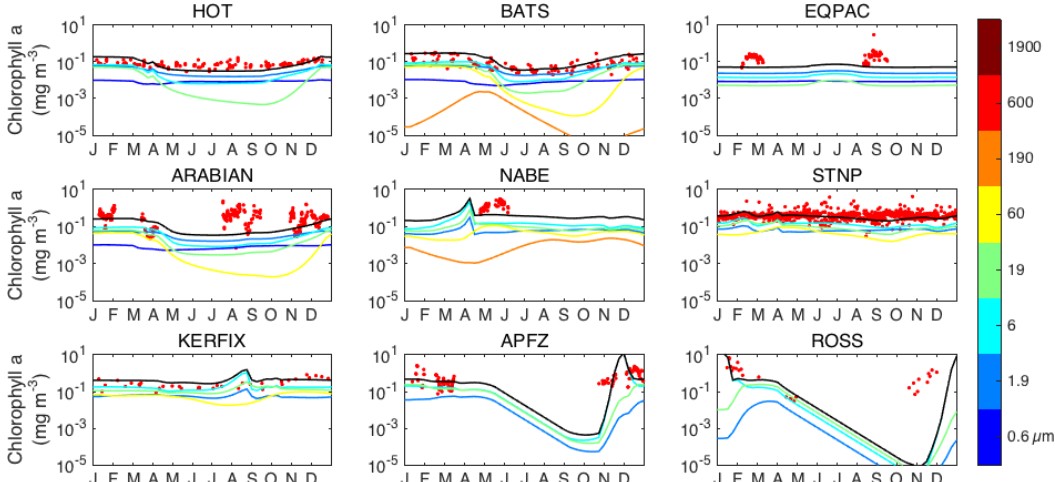

**Figure 17.** Annual cycle of surface chlorophyll *a* at nine JGOFS time-series sites. Red dots indicate climatological observations, while the black lines represents modelled total surface chlorophyll *a*. Coloured lines represent chlorophyll *a* in individual size classes (blue = small, red = large). Locations of the time-series are indicated in Figure 6.



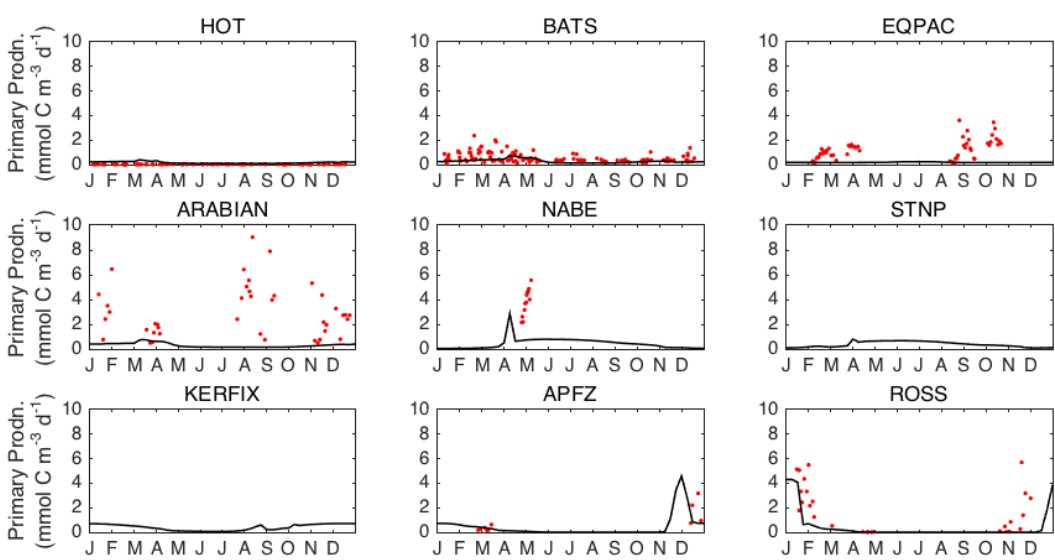

**Figure 18.** Annual cycle of surface primary production at nine JGOFS time-series sites. Red dots indicate climatological observations, while the black lines represents modelled total primary production. Locations of the time-series are indicated in Figure 6.





### 5.2.3 cGENIE vs. EcoGENIE

Figure 19 is a Taylor diagram comparing the two models in terms of their correlation to observa-
tions and their standard deviations, relative to observations. A perfect model would be located at the
middle of the bottom axis, with a correlation coefficient of 1.0 and a normalised standard deviation
of 1.0. The closer a model is to this ideal point, the better a representation of the data it provides.
Figure 19 shows that EcoGENIE is located further from the ideal point than cGENIE, in terms of
oxygen, alkalinity, phosphate, and DIC. The new model seems to provide a universally worse repre-
sentation of global ocean biogeochemistry. This is perhaps not surprising, given that the BIOGEM
component of cGENIE has at various times been systematically tuned to match the observation data
(e.g. Ridgwell et al., 2007a; Ridgwell and Death, in prep.). EcoGENIE has not yet been optimised
in this way.

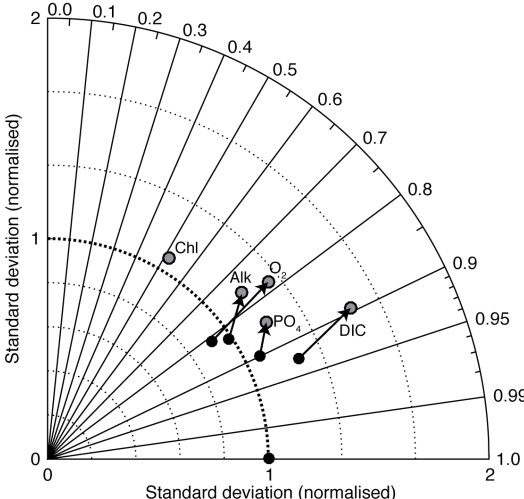

**Figure 19.** Taylor diagram comparing cGENIE and EcoGENIE to annual mean observation fields.





## 6 Discussion

The marine ecosystem is a central component of the Earth system, harnessing solar energy to sustain the biogeochemical cycling of elements between dissolved inorganic nutrients, living biomass and decaying organic matter. The interaction of these components with the global carbon cycle is critical to our interpretation of past, present and future climates, and has motivated the development of a wide range of models. These can be placed on a spectrum of increasing complexity, from simple and
computationally efficient box models to fully coupled Earth system models with extremely large computational overheads.

cGENIE is a model of intermediate complexity on this spectrum. It has been designed to allow rapid model evaluation while at the same time retaining somewhat realistic global dynamics that facilitate comparison with observations. With this goal in mind, the biological pump was parame-
terised as a simple vertical flux defined as a function of environmental conditions (Ridgwell et al., 2007a). This simplicity is well suited to questions concerning the interactions of marine biogeochemistry and climate, but at the same time precludes any investigation of the role of ecological interactions with the broader Earth system.

Here we have presented an ecological extension to cGENIE that opens up this area of investiga-
tion. EcoGENIE is rooted in size-dependent physiological and ecological constraints (Ward et al., 2012). The ecophysiological parameters are relatively well constrained by observations, even in comparison to simpler ecosystem models that are based on much more aggregated functional groups (Anderson, 2005; Litchman et al., 2007). The size-based formulation has the additional benefit of linking directly to functional aspects of the ecosystem, such as food-web structure and particle sink-
ing (Ward and Follows, 2016).

The aim of this paper is to provide a detailed description of the new ecological component. It is clear from Figure 19 that the switch from the parameterised biological pump to the explicit ecological model has led to a deterioration in the overall ability of cGENIE to reproduce the global distributions of important biogeochemical tracers. This is an acceptable outcome, as our goal here is simply to
provide a full description of the new model. Given that the original model was calibrated to the observations in question (Ridgwell et al., 2007a), that process will need to be repeated for the new model before any sort of objective comparison can be made. We also note that EcoGENIE is still capable of reproducing approximately 90% of the global variability in DIC, more than 70% for phosphate, oxygen and alkalinity, and more than 50% for surface chlorophyll.

Despite a slight overall deterioration in terms of model-observation misfit, the biogeochemical components of the model retain the key features that should be expected. At the same time, the ecological community conforms to expectations in terms of standing stocks and fluxes, both in terms of large-scale spatial distributions, and the seasonal cycles at specific locations. Overall patterns of community structure and physiological limitation also follow expectations based on observations
and theory.





As presented, the model is limited to three limiting resources (light, phosphorus, and iron) and two plankton functional types (phytoplankton and zooplankton). We have written the model equations and code to facilitate the extension of the model to include additional components. In particular, the model capabilities can be extended by enabling silicon and nitrogen limitation, leveraging the silicon

and nitrogen cycles already present in BIOGEM (Monteiro et al., 2012). Adding these nutrients will enable the addition of diatoms and diazotrophs, which are both likely to be important factors affecting the strength of the long-term biological pump (Tyrrell, 1999; Armstrong et al., 2002).

## 7 Code availability

The model code and user instructions can be found at http://www.seao2.info/mycgenie.html.

SVN revision 9982.

*Acknowledgements.* This work was supported by the European Research Council 'PALEOGENiE' project (ERC-2013-CoG-617313). BAW thanks the Marine Systems Modelling group at the National Oceanography Centre, Southampton.





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
