# Peer review of "EcoGEnIE 0.2: Plankton Ecology in the cGEnIE Earth system model"

_Geoscientific Model Development, 2017_

## Short Comment (SC1) · 17 Nov 2017

In your section 4.2 (observations) you state that you use DIC and alkalinity data from the Global Ocean Data Analysis Project (GLODAP). No reference is given to these data, and I am uncertain which observational data have actually been used. There are currently two versions of GLODAP: GLODAPv1.1 is the original, published in 2004 and referenced as Key et al (2004); GLODAPv2 is the updated version, published in 2016 and referenced as either Olsen et al (2016) if you used the discrete data or Lauvset et al (2016) if you used the mapped/gridded data. All references are given below. Given that in your Figure 3 it looks like there are no DIC and alkalinity data in the Arctic Ocean, I assume that you have used the original GLODAP (Key et al, 2014) which had no data in that ocean region. Regardless of which observational data are used you have to

add the correct reference and preferrably also link to the actual data. I would, however, strongly encourage you to use the updated GLODAPv2 as it has far more data and underwent improved data quality control measures. It is an improved observational data product and therefore more appropriate for your model-data comparison. GLODAPv2 can be downloaded here: https://www.nodc.noaa.gov/ocads/oceans/GLODAPv2/, and you will also find links to the publications describing the data at that site. I also notice that there is no proper reference to the WOA data used. Appropriate references can be found here: https://www.nodc.noaa.gov/OC5/indprod.html

References: Lauvset, S. K., et al. (2016), A new global interior ocean mapped climatology: the $1\times1$ GLODAP version 2, Earth Syst. Sci. Data, 8(2), 325-340, doi:10.5194/essd-8-325-2016.

Olsen, A., et al. (2016), The Global Ocean Data Analysis Project version 2 (GLODAPv2) – an internally consistent data product for the world ocean, Earth Syst. Sci. Data, 8(2), 297-323, doi:10.5194/essd-8-297-2016.

Key, R. M., A. Kozyr, C. L. Sabine, K. Lee, R. Wanninkhof, J. L. Bullister, R. A. Feely, F. J. Millero, C. Mordy, and T. H. Peng (2004), A global ocean carbon climatology: Results from Global Data Analysis Project (GLODAP), Global Biogeochemical Cycles, 18(4), doi:Artn Gb4031 10.1029/2004gb002247.
* * *

---

## Referee Comment (RC1) · E. Buitenhuis (Referee) · 13 Dec 2017

This manuscript is a generally clear description of an extended ocean biogeochemical model in the GENIE EMIC. However, a number of equations are wrong, so I recommend correcting these, rerunning the model, updating the figures, and if needed the text and conclusions. While this amounts to major revisions, I am hopeful that a revised manuscript would be acceptable.

Major comments: Line 244: "Finally organic matter (D) is made up of K size classes of organic matter, each containing i d organic nutrient element pools. (Note that strictly speaking, detrital organic matter is not explicitly resolved as a state variable in ECO-GEM as we currently only resolve the production of organic matter, which is passed to

BIO- GEM and held there as a state variable. As a consequence, there is no grazing on detrital organic matter in the current configuration of EcoGENIE. We include a description of D and its relationships here for completeness and for convenience of notation." This is in fact a really inconvenient notation, because it obscures what happens in the model. See Line 452 below.

Line 294: "The size of the quota increases with [. . .] the loss of carbon." First, this statement is not true, because in Section 3.2.7 it is pointed out that grazing loss does not affect stoichiometry, which is the correct thing to do. Secondly, Ikeda et al. (2001, DOI 10.1007/s002270100608) show that the stoichiometry of respiration is undistinguishable from the stoichiometry of biomass as well. This incorrect representation would lead to quota that are sometimes in excess of $Q_{max}$, which would give unrealistic artefacts in the nutrient cycling and potentially a violation of mass balance, and should therefore be corrected.

Line 301: This incorrect equation also appears in Geider et al. (1998, Limnol. Oceanogr. 43:679), although it is given here without attribution. Please use the correct equation from Morel (1987, J. Phycol. 23:137) max = himax – (himax – lomax)*(Q-Qmin)/(Qmax-Qmin)

Line 303: The appearance of $\gamma Fe$ in the denominator of this equation is incorrect. It would make Chl synthesis increase as cells run out of iron, when in fact Chl:C decreases at low iron (Sunda and Huntsman 1997, Nature 390:389). A photosynthesis model that reproduces this iron limitation effect is given in Buitenhuis and Geider (2010, Limnol. Oceanogr. 55:714)

Section 3.2.6 uses several words that have physiological meanings (limitation term, half saturation, inhibition) in a section describing light attenuation. If these sentences in fact deal with $\alpha$ (it would help to rename this to $\alpha Chl$), then it should be moved to Section 3.4.3. If it deals with light attenuation, it should be made clear how kChl is derived.

Line 343 : "length scale of 20 m" Is this used to calculate kw or the average value of ktot?

Line 345: "At the ocean surface" This would be a logical sentence to start the section.

Between Line 452 and 455 D changes from 6 state variables in ECOGEM to 2 (C contents) in BIOGEM. Please explain what happens to the organic nutrient concentrations.

Line 533: It would make more sense to change e.g. the range between Qmin and Qmax, the partitioning between POM and DOM and the decay of POM with depth, which have much more uncertainty than the unrealistic choice noted in Line 294.

Section 3.2.9: See comment on Line 294.

Figure 5 and Line 595: It is confusing to speak of POC production when there is no state variable for POC, and it leads to confusion with primary production. It would be easier to understand to speak of POC flux. Given the central importance of POC flux for air-sea CO2 flux and nutrient distributions, I suggest comparing it to observations (Schlitzer (2004), J. of Oceanography 60:53-62, https://lred.uea.ac.uk/web/green-ocean/data ) and including these in Figures 3 and 19.

Line 605: Rather than change ECOGEM to reproduce an arbitrary result in BIOGEM, it would be much more helpful to compare the CaCO3 export to observations (Lee (2001) Limnol. Oceanogr. 46: 1287–1297) and adjust the model to reproduce that.

Line 617: "total oceanic DIC inventory increased by just under 2% from 0.299 mol C" This makes no sense. The total oceanic DIC inventory is $\sim$3.3 Examol.

Line 652 and Figure 17: "The model predicts higher chlorophyll concentrations in the Southern Ocean" Figure 17 is inadequate to decide whether this is a reasonable comment to make. Please have the y-axis range from 10-2 to 10 (values between 10-5 and 10-2 are insignificant), put the station names inside the panels, so that the panels can be made higher, and include the satellite chl in the figure. If that shows the in situ measurements span the satellite estimates, delete the Dierssen reference and rewrite

this to reflect the findings of Le Quere et al. (2016, doi:10.5194/bg-13-4111-2016), that models underestimate SO chl because they underrepresent macrozooplankton. Also, after correcting the error on line 303, this may improve/decrease SO chl.

Figure 18 needs to be described.

Line 737: "the ecological community conforms to expectations in terms of standing stocks" This has not been shown. Comparison to Buitenhuis et al. (2013, doi:10.5194/essd-5-227-2013) would test this statement. Given the different definition of plankton groups, comparison could be made to Fig. 5a.

Minor comments: Line 85: for the how -> for how Line 143: in terms its -> in terms of its Line 162: modularised -> modular Line 176: a greater intention to explore long timescale -> the intention to explore longer timescale Line 262: the the -> the Line 363: level the -> level of the Line 421: The O2:C ratio is in fact >1. Anderson and Sarmiento (1994) find it's ~170/117=1.45, so even 138/106 would be quite low, and it would be helpful to justify it. Line 439: used equations -> used in equations Line 592: in tropical -> in the tropical Line 636: Figures 12 and 13 we -> In Figures 12 and 13 we Line 687: Figure 18 -> Figure 17 Line 690: is probably too low -> is too low

---

## Short Comment (SC2) · 13 Dec 2017

As explained in https://www.geoscientific-model-development.net/about/manuscript_types.html GMD is encouraging that authors upload the program code of models (including relevant data sets) as a supplement or make the code and data accessible through a DOI (digital object identifier) for the exact model version described in the paper. In case your institution does not provide the possibility to make data accessible through a DOI you may consider other providers (eg. zenodo.org of CERN) to create a DOI.

Lutz Gross GMD Executive Editor

2017.

---

## Referee Comment (RC2) · Anonymous Referee #2 · 17 Jan 2018

Ward et al. present a new, size-based, marine ecosystem module for the EMIC "GENIE", called "ECOGEM", that is intended to replace the simpler module "BIOGEM". They compare the results of two long-term simulations with these different ecosystem modules.

**General Comments**

The manuscript is generally well written; the ECOGEM equations are presented in a comprehensible way. Since this module has been used in a previous study (Ward et al. 2012), I will only comment on the specific use of ECOGEM in GENIE. Specifically, I am missing a critical discussion concerning ecosystem complexity versus simplifications in GENIE and possible problems related to light attenuation, export production (no

prognostic variable for POC) and the neglect of physical transport of the ecosystem variables.

In addition, the results section must be improved as there are several shortcomings (see below); some figures are poorly explained.

**Specific Comments**

- title, line 158: please use a consistent terminology: either EcoGE**n**IE or EcoGE**N**IE

- line 21: please rephrase: fisheries is not "life in the ocean"

- line 25: since the reference is the latest but not the most common or original work, please use at least **e.g.** Hülse et al., 2017

- Figure 1: a similar figure for cGENIE and not only EcoGENIE would be helpful to immediately see the differences in complexity

- sections 3.2.5 Photoacclimation/3.2.6 Light attenuation: Please explicitly state that "photoacclimation" will not be relevant in this current ESM setup. The light attenuation in "GENIE" is overly simplified by assuming an *average* irradiance for the entire surface mixed layer and zero below. The idea to introduce a variable C:Chl ratio is mainly to allow for the development of subsurface chlorophyll maxima that do not correspond to phytoplankton biomass (carbon) maxima. Since the model resolution is too coarse and mean light levels are assumed, the C:Chl-ratio will not vary with depth.

- lines 406/407: In my experience a minimum concentration of 1 x$10^{-6}$ mmol C m$^{-3}$ is high and will affect the results significantly; the variability and signals (like

extinction) that might become relevant on longer time scales will be smeared out. A smaller value should be used at least for future studies.

- section 3.3.3 Dissolved organic matter: please explicitly state that export production within and below the mixed layer is the same (otherwise the figure caption in Figure 5 is confusing).

- section 4.2 Observations: the references for all observations must be properly provided (e.g. WOA09 is not sufficient)

- section 5 Results: the entire section is presented in a very sloppy way. A few more explanations about why differences occur between the results of both model configurations or between model and observations are necessary. Please also provide the units for *all* quantities in *all* figures!

- section 5.1.1 Global surface values: there is general agreement that primary production in the Southern Ocean (the largest HNLC area) is limited by iron, which explains the high macronutrient (e.g. phosphate) concentrations. ESMs generally overestimate the iron concentrations and thus nutrient uptake in the SO. Although the phosphate concentrations in the SO (Fig. 3) are difficult to identify, it seems to me that this is the case here, too. Is it true?

- line 683: please be more specific. The general statement "Iron limitation in high latitude regions" is wrong. As far as I can deduce from Figure 16, iron limitation occurs mainly in the Southern Ocean and the western part of the North Subarctic Pacific Ocean.

- line 711: "costs" should be used here instead of "overheads".

- all Figures showing spatial maps: what does the number 10000 on the North American continent refer to?

---

## Author Comment (AC3) · 4 May 2018

We thank the commenter for taking the time to point out these issues with the observational data. We have updated the observed DIC and Alkalinity, gridding the GLODAPv2 data onto the GEnIE grid, and citing Olsen et al. 2016. We also now include a reference for the WOA (2009) data (Garcia et al. 2010).

---

## Author Comment (AC4) · 12 May 2018

We are currently in the process of migrating the code from svn to git. A DOI for this will be provided with the revised manuscript.
* * *

---

## Author Response (AR1)

This manuscript is a generally clear description of an extended ocean biogeochemical model in the GENIE EMIC. However, a number of equations are wrong, so I recom- mend correcting these, rerunning the model, updating the figures, and if needed the text and conclusions. While this amounts to major revisions, I am hopeful that a revised manuscript would be acceptable.

**We would like to thank the reviewer for a very thoughtful and constructive review.**

Major comments: Line 244: "Finally organic matter (D) is made up of K size classes of organic matter, each containing  $i_d$  organic nutrient element pools. (Note that strictly speaking, detrital organic matter is not explicitly resolved as a state variable in ECOGEM as we currently only resolve the production of organic matter, which is passed to BIOGEM and held there as a state variable. As a consequence, there is no grazing on detrital organic matter in the current configuration of EcoGENIE. We include a description of D and its relationships here for completeness and for convenience of notation." This is in fact a really inconvenient notation, because it obscures what happens in the model. See Line 452 below.

**Please see our response to the comment relating to lines 452-455.**

Line 294: "The size of the quota increases with [. . .] the loss of carbon." First, this statement is not true, because in Section 3.2.7 it is pointed out that grazing loss does not affect stoichiometry, which is the correct thing to do.

We have now removed the preferential loss of carbon (through respiration) in response to a comment below. As such, we have also removed this statement, which is no longer applies to the updated model.

Secondly, Ikeda et al. (2001, DOI 10.1007/s002270100608) show that the stoichiometry of respiration is undistinguishable from the stoichiometry of biomass as well. This incorrect representation would lead to quota that are sometimes in excess of Qmax, which would give unrealistic artefacts in the nutrient cycling and potentially a violation of mass balance, and should therefore be corrected.

We have removed the independent loss of carbon from the model, as it can indeed violate mass balance in some conditions.

Line 301: This incorrect equation also appears in Geider et al. (1998, Limnol. Oceanogr. 43:679), although it is given here without attribution. Please use the correct equation from Morel (1987, J. Phycol. 23:137) max = himax – (himax – lomax)\*(Q- Qmin)/(Qmax-Qmin)

As is the case for all models, both the Geider and Morel formulations have their issues. It is misleading to refer to either one as "correct" or "incorrect". The Morel model, for example predicts that Q=Qmax when  $\mu$ = $\mu$ max. This prediction is clearly refuted for non-limiting nutrients by Elrifi & Turpin (1985, J. Phycol., 21, 592–602). For the sake of maintaining consistency with Ward et al. (2012), we have chosen to retain the Geider et al. (2007) formulation.

Line 303: The appearance of  $\gamma$ Fe in the denominator of this equation is incorrect. It would make Chl synthesis increase as cells run out of iron, when in fact Chl:C decreases at low iron (Sunda and Huntsman 1997, Nature 390:389). A photosynthesis model that reproduces this iron limitation effect is given in Buitenhuis and Geider (2010, Limnol. Oceanogr. 55:714)

Assuming the reviewer means equation 13, The appearance of  $\gamma$ Fe in the denominator does not imply that ChI synthesis increases as cells become Fe limited, because, when Fe is limiting,  $\gamma$ Fe also appears the numerator (via Pc and Psat; Equations 9 and 11).

Section 3.2.6 uses several words that have physiological meanings (limitation term, half saturation, inhibition) in a section describing light attenuation. If these sentences in fact deal with  $\alpha$  (it would help to rename this to  $\alpha$ ChI), then it should be moved to Section 3.4.3. If it deals with light attenuation, it should be made clear how kChI is derived.

We have rearranged the text accordingly.

Line 343 : "length scale of 20 m" Is this used to calculate kw or the average value of ktot?

BIOGEM doesn't represent ChI, so water attenuates light with a constant optical depth. We have modified the text to clarify this.

"In both BIOGEM and ECOGEM, the incoming shortwave solar radiation intensity is taken from the climate component in cGEnIE and varies seasonally (Edwards and Marsh, 2005b; Marsh et al., 2011). However, ECOGEM uses a slightly more complex light attenuation scheme than BIOGEM, which simply calculates a mean solar (shortwave) irradiance averaged over the depth of the surface layer, assuming a clear-water light attenuation scale of 20 m (Doney et al., 2006)."

Line 345: "At the ocean surface" This would be a logical sentence to start the section.

We have rearranged the text to provide a more logical order to the sentence.

Between Line 452 and 455 D changes from 6 state variables in ECOGEM to 2 (C contents) in BIOGEM. Please explain what happens to the organic nutrient concentrations.

The ambiguity probably arose through our use of the singular in reference to the POM and DOM state-variable/flux vectors (each corresponding to three ECOGEM state-variables). We have changed the text to make it clear that there are 3 DOM state-variables and three POM fluxes in BIOGEM.

"The dissolved organic matter vector (D1) includes three explicit tracers that are transported by the ocean circulation model and are degraded back to their constituent nutrients with a fixed turnover time of  $\lambda$  (= 0.5 years). Particulate organic matter (POM) is not represented with explicit state vari- ables in either ECOGEM or BIOGEM. Instead, its implicit production in the surface layer (and the corresponding export below the surface layer) is given by..."

Line 533: It would make more sense to change e.g. the range between Qmin and Qmax, the partitioning between POM and DOM and the decay of POM with depth, which have much more uncertainty than the unrealistic choice noted in Line 294.

We have increased QminP (i.e. decreased max biomass C:P ratio) to compensate for the removal of C respiration.

Section 3.2.9: See comment on Line 294.

We have addressed the choice at line 294 above.

Figure 5 and Line 595: It is confusing to speak of POC production when there is no state variable for POC, and it leads to confusion with primary production. It would be easier to understand to speak of POC flux. Given the central importance of POC flux for air-sea CO2 flux and nutrient distributions, I suggest comparing it to observations (Schlitzer (2004), J. of Oceanography 60:53-62, https://lred.uea.ac.uk/web/green-ocean/data) and including these in Figures 3 and 19.

We have changed the text to describe POC flux rather than production.

"The relative proportions in which these elements and compounds are exported from the surface ocean are regulated by the stoichiometry of biological production. In cGEnIE (BIOGEM), carbon 595 and phosphorus production are rigidly coupled through a fixed ratio of 106:1, while POFe:POC and CaCO3:POC production ratios are regulated as a function of environmental conditions. In ecoGEnIE (ECOGEM), phosphorus, iron and carbon production are all decoupled through the flexible quota physiology, which depends on both environmental conditions, and the status of the food-web. Only CaCO3:POC production ratios are regulated via the same mechanism in the two models (although 600 we decreased the average CaCO3:POC ratio in ECOGEM to compensate for the elevated POC production relative to POP)."

We prefer not to use the Schlitzer POC flux dataset. It is based on data assimilation exercise in the North Pacific, and it is not clear how well it extrapolates to the global scale. Indeed, global estimates for vertical POC are still highly uncertain and even contradictory (see for example the discrepancy between Henson et al. doi:10.1029/2011GL046735, 2011 and Marsay et al. 10.1073/pnas.1415311112 2015), so we would prefer not to use these data as a benchmark of model performance.

Line 605: Rather than change ECOGEM to reproduce an arbitrary result in BIOGEM, it would be much more helpful to compare the CaCO3 export to observations (Lee (2001) Limnol. Oceanogr. 46: 1287–1297) and adjust the model to reproduce that.

The BIOGEM result was not arbitrary. It was from a model systematically calibrated to global phosphate and alkalinity measurements. An important aspect of the work here is traceably distinguishing the performance of ECOGEM and BIOGEM, so it is important to consistently evaluate the former against the latter.

Line 617: "total oceanic DIC inventory increased by just under 2% from 0.299 mol C" This makes no sense. The total oceanic DIC inventory is ~3.3 Examol.

Thanks for pointing this out. The 'Exa' prefix was omitted in error.

Line 652 and Figure 17: "The model predicts higher chlorophyll concentrations in the Southern Ocean" Figure 17 is inadequate to decide whether this is a reasonable comment to make.

This comment was in reference to Figure 14, which shows higher model chlorophyll concentrations in the Southern Ocean, relative to the SeaWiFS data.

Please have the y-axis range from 10-2 to 10 (values between 10-5 and 10-2 are insignificant), put the station names inside the panels, so that the panels can be made higher, and include the satellite chl in the figure.

The y-axis range was chosen to show the dynamic range of the model. Values < 1e-2 are indeed insignificant in the observations, but low winter values are an important component of the model dynamics. As such, we feel it is important to retain the y-axis range of 1e-5 to 1e1.

**We have added SeaWiFs chlorophyll to the figure.**

If that shows the in situ measurements span the satellite estimates, delete the Dierssen reference and rewrite this to reflect the findings of Le Quere et al. (2016, doi:10.5194/bg-13-4111-2016), that models underestimate SO chl because they underrepresent macrozooplankton. Also, after correcting the error on line 303, this may improve/decrease SO chl.

We have plotted SeaWiFS data at the three Southern Ocean sites. The satellite data do show a tendency to underestimate in situ observations on the Southern Ocean. We have therefore retained the Dierssen reference.

**Figure 18 needs to be described.**

**We have added a description.**

"The seasonal cycles of primary production in the surface layer are compared to time-series observations in Figure 18. As also indicated in Figure 14, the spatial variance in modelled primary production is too low, with primary production overestimated at the most oligotrophic site (HOT) and typically underestimated at the most productive sites (esp. the equatorial Pacific, NABE and the Ross Sea). In contrast to the lack of spatial variability, the model exhibits significant seasonal variation, often in excess of the observed variability (at those sites where the seasonal cycle is well resolved)."

Line 737: "the ecological community conforms to expectations in terms of standing stocks" This has not been shown. Comparison to Buitenhuis et al. (2013, doi:10.5194/essd-5-227-2013) would test this statement. Given the different definition of plankton groups, comparison could be made to Fig. 5a.

**We have corrected this statement to read "the ecological community conforms to expectations in terms of standing stocks and fluxes, both in terms of large-scale spatial distributions, and the seasonal cycles at specific locations (Figures 14 and 17)".**

Minor comments: Line 85: for the how -> for how Line 143: in terms its -> in terms of its Line 162: modularised -> modular Line 176: a greater intention to explore long timescale -> the intention to explore longer timescale Line 262: the the -> the Line 363: level the -> level of the Line 421: The O2:C ratio is in fact >1. Anderson and Sarmiento (1994) find it's ~170/117=1.45, so even 138/106 would be quite low, and it would be helpful to justify it. Line 439: used equations -> used in equations Line 592: in tropical -> in the tropical Line 636: Figures 12 and 13 we -> In Figures 12 and 13 we Line 687: Figure 18 -> Figure 17 Line 690: is probably too low -> is too low

We will correct all these errors in the resubmitted manuscript. Note that the O2:CO2 ratio was inverted in error. This has been corrected.

Ward et al. present a new, size-based, marine ecosystem module for the EMIC "GENIE", called "ECOGEM", that is intended to replace the simpler module "BIOGEM". They compare the results of two long-term simulations with these different ecosystem modules.

We would like to thank the reviewer for a very thoughtful and constructive review.

**General Comments**

The manuscript is generally well written; the ECOGEM equations are presented in a comprehensible way. Since this module has been used in a previous study (Ward et al. 2012), I will only comment on the specific use of ECOGEM in GENIE. Specifically, I am missing a critical discussion concerning ecosystem complexity versus simplifications in GENIE and possible problems related to light attenuation, export production (no prognostic variable for POC) and the neglect of physical transport of the ecosystem variables.

In addition, the results section must be improved as there are several shortcomings (see below); some figures are poorly explained.

**Specific Comments**

• title, line 158: please use a consistent terminology: either EcoGEnIE or EcoGENIE

We have changed to GEnIE and EcoGEnIE throughout.

• line 21: please rephrase: fisheries is not "life in the ocean"

Changed to "support almost all life in the ocean, including the fish stocks that provide essential nutrition to more than half the human population".

• line 25: since the reference is the latest but not the most common or original work, please use at least **e.g.** Hülse et al., 2017

**Changed.**

• Figure 1: a similar figure for cGENIE and not only EcoGENIE would be helpful to immediately see the differences in complexity

The figure currently includes the BIOGEM module (cGEnIE & EcoGEnIE) and the ECOGEM module (EcoGEnIE only). This think this should be sufficient for understanding the relationship between the two models.

 sections 3.2.5 Photoacclimation/3.2.6 Light attenuation: Please explicitly state that "photoacclimation" will not be relevant in this current ESM setup. The light attenuation in "GENIE" is overly simplified by assuming an *average* irradiance for the entire surface mixed layer and zero below. The idea to introduce a variable C:Chl ratio is mainly to allow for the development of subsurface chlorophyll maxima that do not correspond to phytoplankton biomass (carbon) maxima. Since the model resolution is too coarse and mean light levels are assumed, the C:Chl- ratio will not vary with depth.

The C:Chl ratio also varies horizontally, as a function of PAR and nutrient availability. We included it to make comparisons to satellite data more meaningful.

lines 406/407: In my experience a minimum concentration of  $1 \times 10^{-6}$  mmol C m-3 is high and will affect the results significantly; the variability and signals (like extinction) that might become relevant on longer time scales will be smeared out. A smaller value should be used at least for future studies.

**Thanks for pointing this out.**

• section 3.3.3 Dissolved organic matter: please explicitly state that export production within and below the mixed layer is the same (otherwise the figure caption in Figure 5 is confusing).

Changed to "*implicit production in the surface layer (and the corresponding export below the surface layer) is given by…*".

• section 4.2 Observations: the references for all observations must be properly provided (e.g. WOA09 is not sufficient)

We now cite references for the GLODAPv2 and WOA data used. We also acknowledge the source of the SeaWiFs Chl data in the acknowledgements.

• section 5 Results: the entire section is presented in a very sloppy way. A few more explanations about why differences occur between the results of both model configurations or between model and observations are necessary. Please also provide the units for *all* quantities in *all* figures!

Units are now included for all figures. We also provide a more complete description of the results (e.g. description of Figure 18).

section 5.1.1 Global surface values: there is general agreement that primary
production in the Southern Ocean (the largest HNLC area) is limited by iron, which
explains the high macronutrient (e.g. phosphate) concentrations. ESMs generally
overestimate the iron concentrations and thus nutrient uptake in the SO. Although
the phosphate concentrations in the SO (Fig. 3) are difficult to identify, it seems to
me that this is the case here, too. Is it true?

The high surface PO4 concentrations in the SO are likely a consequence of low Fe, low irradiance (deep mixing) and cold temperatures.

• line 683: please be more specific. The general statement "Iron limitation in high latitude regions" is wrong. As far as I can deduce from Figure 16, iron limitation occurs mainly in the Southern Ocean and the western part of the North Subarctic Pacific Ocean.

Fe limitation is clearly seen in all high latitude regions (especially among the larger phytoplankton size classes). We have adjusted the text to highlight this in Figure 16.

*"Iron limitation dominates in high latitude regions, especially among larger size classes. Among these larger groups, the upwelling zones appear to be characterised by iron-phosphorus co-limitation."*

• line 711: "costs" should be used here instead of "overheads".

Changed.

• all Figures showing spatial maps: what does the number 10000 on the North American continent refer to?

This was the model integration year. It has been removed from all figures.

Manuscript prepared for Geosci. Model Dev. with version 2015/09/17 7.94 Copernicus papers of the LATEX class copernicus.cls. Date: 6 July 2018

**EcoGEnIE 0.10.2**: Plankton Ecology in the **cGENIE cGEnIE** Earth system model**

Ben A. Ward1,2,3, Jamie D. Wilson1, Ros M. Death1, Fanny M. Monteiro1, Andrew Yool2, and Andy Ridgwell1,4

[revised manuscript text omitted]

$$\quad P_{j,C} = \gamma_{j,I} P_{j,C}^{\text{sat}} \tag{11}$$

Net carbon uptake is given by

$$V_{j,C} = P_{j,C} - \xi \cdot V_{j,P} \tag{12}$$

With the second term accounting for the metabolic cost of biosynthesis ( $\xi$ ). This parameter was originally defined as a loss of carbon as a fraction of nitrogen uptake (Geider et al.) [1998). We define it here relative to phosphate uptake, using a fixed N:P ratio of 16.

330

335

**3.2.5 Photoacclimation**

The chlorophyll-to-carbon ratio is regulated as the cell attempts to balance the rate of light capture by chlorophyll with the maximum potential (i.e. light-replete) rate of carbon fixation. Depending on this ratio, a certain fraction of newly assimilated phosphorus is diverted to the synthesis of new chlorophyll *a*,

Pro

$$\rho_{j,\text{Chl}} = \theta_{\text{P}}^{\max} \frac{I_{j,\text{C}}}{\alpha \cdot \gamma_{j,\text{Fe}} \cdot Q_{j,\text{Chl}} \cdot I}$$
(13)

Here  $\rho_{j,\text{Chl}}$  is the amount of chlorophyll *a* that is synthesised for every mmol of phosphorus assimilated (mg Chl (mmol P)-1) with  $\theta_{P}^{\text{max}}$  representing the maximum ratio (again converting from the nitrogen based units of Geider et al., [1998] with a fixed N:P ratio of 16). If phosphorus is assimilated at each an encode  $V_{P}$  (mmol P)-1 d-1 d-1) then the each an encode  $V_{P}$  (mmol P)-1 d-1 d-1) then the each and the encoded of the e

340 at carbon specific rate  $V_{j,P}$  (mmol P (mmol C)-1 d-1), then the carbon specific rate of chlorophyll *a* synthesis (mg chl (mmol C)-1 d-1) is

$$V_{j,\text{Chl}} = \rho_{j,\text{Chl}} \cdot V_{j,\text{P}} \tag{14}$$

**3.2.6 Light attenuation**

ECOGEM In both BIOGEM and ECOGEM, the incoming shortwave solar radiation intensity is
taken from the climate component in cGEnIE and varies seasonally (Edwards and Marsh 2005b) Marsh et al. 2011)
. However, ECOGEM uses a slightly more complex light attenuation scheme than BIOGEM, which simply calculates a mean solar (shortwave) irradiance averaged over the depth of the surface layer, and assuming a length assuming a clear-water light attenuation scale of 20 m over which light decays (Doney et al. 2006). BIOGEM then takes this mean irradiance and applies a Michaelis-Menten like
limitation term, assuming a half saturation value for light of 20 W m-2 (Doney et al. 2006). At the ocean surface, the incoming shortwave solar radiation intensity is taken from the climate component in cGENIE and varies seasonally (Edwards and Marsh 2005b) Marsh et al. 2011). (Doney et al. 2006)

$$\quad Chl_{ML} = Chl_{tot} \frac{Z_1}{Z_{ML}} \tag{15}$$

In ECOGEM the light level is calculated as the mean level of photosynthetically available radia-355 tion within a variable mixed layer (with depth calculated according to Kraus and Turner, 1967). We also take into account inhibition of light penetration due to the presence of light absorbing particles and dissolved molecules (Shigsesada and Okubo, 1981). If  $Chl_{tot}$  is the total chlorophyll concentration in the surface layer (of thickness  $Z_1$ ), and  $Z_{ML}$  is the mixed-layer depth, the virtual chlorophyll concentration distributed across the mixed layer is given by

The combined light-attenuation coefficient attributable to both water and the virtual chlorophyll concentration is given by

$$k_{tot} = k_w + k_{chl} \cdot Chl_{ML} \tag{16}$$

For a given level of photosynthetically available radiation at the ocean surface  $(I_0)$ , plankton in the surface grid box experience the average irradiance within the mixed layer, which is given by

$$I = \frac{I_0}{k_{tot}} \frac{1}{Z_{ML}} (1 - e^{(-k_{tot} \cdot Z_{ML})})$$
(17)

**3.2.7 Predation (including both herbivorous and carnivorous interactions)**

Here we define predation simply as the consumption of any living organism, regardless of the trophic level of the organism (i.e. phytoplankton or zooplankton prey).

370 The predator-biomass-specific grazing rate of predator  $(j_{\text{pred}})$  on prey  $(j_{\text{prey}})$  is given by,

$$G_{j_{\text{pred}},j_{\text{prey}},\text{C}} = \gamma_{\text{T}} \underbrace{\underbrace{G_{j_{\text{pred}},\text{C}}^{\text{max}} \cdot \underbrace{\mathcal{F}_{j_{\text{pred}},\text{C}}}_{k_{j_{prey},\text{C}} + \mathcal{F}_{j_{\text{pred}},\text{C}}}}_{\text{overall grazing rate}} \cdot \underbrace{\Phi_{j_{\text{pred}},j_{\text{prey}}}}_{\text{switching}} \cdot \underbrace{(1 - e^{\Lambda \cdot \mathcal{F}_{j_{\text{pred}},\text{C}}})}_{\text{prey refuge}}$$
(18)

where γT is the temperature-dependence, Gmaxjpred,C is the maximum grazing rate, and kjprey,C is the half-saturation concentration for all (available) prey. The overall grazing rate is a function of total
food available to the predator, Fjpred,C. This is given by the product of the prey biomass vector, BC, and the grazing kernel (φ),

$$\mathcal{F}_{\mathrm{C}} = \phi B_{\mathrm{C}} B_{\mathrm{C}}$$
(19)

Note that this equation is written out in matrix form, with the dimensions noted underneath each matrix. Each element of the grazing matrix  $\phi$  is an approximately log-normal function of the predatorto-prey length ratio,  $\vartheta_{j_{\text{pred}},j_{\text{prey}}}$ , with an optimum ratio of  $\vartheta_{\text{opt}}$  and a geometric standard deviation

 $\sigma_{j_{ ext{pred}}}.$

380

$$\phi_{j_{\text{pred}},j_{\text{prey}}} = \exp\left[-\left(\ln\left(\frac{\vartheta_{j_{\text{pred}},j_{\text{prey}}}}{\vartheta_{\text{opt}}}\right)\right)^2 / \left(2\sigma_{j_{\text{pred}}}^2\right)\right]$$
(20)

We also include an optional 'prey-switching' term, such that predators may preferentially attack those prey that are relatively more available (i.e. active switching, s = 2). Alternatively they may attack prey in direct proportion to their availability (i.e. passive switching, s = 1). In the simulations

below we assume active switching.

$$\Phi_{j_{\text{pred}},j_{\text{prey}}} = \frac{(\phi_{j_{\text{pred}},j_{\text{prey}}} B_{j_{\text{prey}},\text{C}})^s}{\sum_{j_{\text{prey}}=1}^J (\phi_{j_{\text{pred}},j_{\text{prey}}} B_{j_{\text{prey}},\text{C}})^s}$$
(21)

Finally, a prey refuge function is incorporated, such that the overall grazing rate is decreased when the availability of all prey ( $\mathcal{F}_{j_{\text{pred}},\text{C}}$ ) is low. The size of the prey refuge is dictated by the coefficient

390  $\Lambda$ . The overall grazing response is calculated on the basis of prey carbon. Grazing losses of other prey elements are simply calculated from their stoichiometric ratio to prey carbon, with different elements assimilated according to the predator's nutritional requirements (see below).

$$G_{j_{\text{pred}},j_{\text{prey}},i_{\text{b}}} = G_{j_{\text{pred}},j_{\text{prey}},\text{C}} \frac{B_{j_{\text{prey}},i_{\text{b}}}}{B_{j_{\text{prey}},\text{C}}}$$
(22)

**3.2.8 Prey assimilation**

395 Prey biomass is assimilated into predator biomass with an efficiency of  $\lambda_{j_{\text{pred}},i_b}$  ( $i_b \neq$  Chl). This has a maximum value of  $\lambda^{\text{max}}$  that is modified according the the quota status of the predator. For elements  $i_b = P$  or Fe, prey biomass is assimilated as a function of the respective predator quota. If the quota is full, the element is not assimilated. If the quota is empty, the element is assimilated with maximum efficiency ( $\lambda^{\text{max}}$ ).

$$\quad \lambda_{j_{\text{pred}},i_b} = \lambda^{\max} Q_{j,i_b}^{\text{stat}} \tag{23}$$

C assimilation is regulated according to the status of the most limiting nutrient element (P or Fe) modified by the same shape-parameter, h, that was applied in Equation 6.

$$Q_{j,i_b}^{\lim} = \left(\frac{Q_{j,i_b} - Q_{j,i_b}^{\min}}{Q_{j,i_b}^{\max} - Q_{j,i_b}^{\min}}\right)^h \tag{24}$$

If both nutrient quotas are full, C is assimilated at the maximum rate. If either are empty, C assimilation is down-regulated until sufficient quantities of the limiting element(s) are acquired.

$$\lambda_{j_{\text{pred}},\text{C}} = \lambda^{\max} \min\left(Q_{j,\text{P}}^{\lim}, Q_{j,\text{Fe}}^{\lim}\
[revised manuscript text omitted]
  | $	heta_{ m N}^{ m max}$                                                 | 48                      | mg Chl $a \pmod{P}^{-1}$                                                       |
| Initial slope of P-I curve         | lpha                                                                    | $3.83 \times 10^{-7}$   | mmol C (mg Chl $a$ ) -1 ( $\mu$ Ein m -2 ) -1 |
| Cost of biosynthesis               | ξ                                                                       | 37.28                   | mmol C (mmol P) $^{-1}$                                                        |
| Grazing                            |                                                                         |                         |                                                                                |
| Optimum predator:prey length ratio | $\vartheta_{\mathrm{opt}}$                                              | 10                      | -                                                                              |
| Geometric s.d. of $\vartheta$      | $\sigma_{ m graz}$                                                      | 2.0                     | -                                                                              |
| Total prey half-saturation         | $k_{ m C}^{ m prey}$                                                    | 5.0                     | $\mathrm{mmol}~\mathrm{C}~\mathrm{m}^{-3}$                                     |
| Maximum assimilation efficiency    | $\lambda^{	ext{max}}$                                                   | 0.7                     | -                                                                              |
| Grazing refuge parameter           | Λ                                                                       | -1                      | $(\text{mmol C } \text{m}^{-3})^{-1}$                                          |
| Active switching parameter         | 8                                                                       | 2                       | -                                                                              |
| Assimilation shape parameter       | h                                                                       | 0.1                     | -                                                                              |
| Other loss terms                   |                                                                         |                         |                                                                                |
| Plankton mortality                 | m                                                                       | 0.05                    | $d^{-1}$                                                                       |
| Plankton respiration               | $r_{i_b = \mathrm{DIC}} 0.05 \mathrm{d}^{-1} r_{i_b \neq \mathrm{DIC}}$ | 0                       | $d^{-1}$                                                                       |
| Partitioning of organic matter     |                                                                         |                         |                                                                                |
| Fraction to DOM                    | eta                                                                     | 0.66                    | -                                                                              |
| Light attenuation                  |                                                                         |                         |                                                                                |
| Light attenuation by water         | $k_{ m w}$                                                              | 0.04                    | $m^{-1}$                                                                       |
| Light attenuation by chlorophyll   | $k_{ m Chl}$                                                            | 0.03                    | $m^{-1}(mg \ Chl)^{-1}$                                                        |
|                                    |                                                                         |                         |                                                                                |

**Table 4. Size-independent model parameters.**

[revised manuscript text omitted]

- In Figure we show the annual mean rate of particulate organic matter production in the surface layer, and the relative differences between ECOGEM and BIOGEM. In comparison to eGENIE, EcoGENIE cGEnIE, EcoGEnIE shows elevated POC production in all regions. Production of CaCO3 is globally less variable in EcoGENIE than cGENIE than cGEnIE, with notable higher fluxes in the oligotrophic gyres and polar regions.
- 610 The relative proportions in which these elements and compounds are exported from the surface ocean are regulated by the stoichiometry of biological production. In eGENIE\_cGEnIE (BIO-GEM), carbon and phosphorus production are rigidly coupled through a fixed ratio of 106:1, while POFe:POC and CaCO3:POC production export flux ratios are regulated as a function of environmental conditions. In ecoGENIE ecoGEnIE (ECOGEM), phosphorus, iron and carbon production

615 fluxes are all decoupled through the flexible quota physiology, which depends on both environmental conditions, and the status of the food-web. Only CaCO3:POC production\_flux ratios are regulated via the same mechanism in the two models (although we decreased the average CaCO3:POC ratio in ECOGEM to compensate for the elevated POC production relative to POP).

---

## Author Response (AR2)

Dear Paul,

Thanks for your comments. In response, I think we have addressed the general concerns of Reviewer 2 (although perhaps not so well in the response). I respond to the reviewers general comments below, with their comments in bold.

**I am missing a critical discussion concerning ecosystem complexity versus simplifications in GENIE and possible problems related to light attenuation, export production (no prognostic variable for POC) and the neglect of physical transport of the ecosystem variables.**

**critical discussion concerning ecosystem complexity versus simplifications in GENIE**

While this comment is quite vague, and I think we have allocated a reasonable amount of text to the issue of complexity. In particular, there is a fairly long section of the Introduction that tackles the issue of Eco-BGC model complexity and our motivation for developing a physically simple yet ecologically complex model.This is also noted in the abstract "The increased capabilities of EcoGENIE in this regard will enable future exploration of the ecological community on much longer timescales than have previously been examined in global ocean ecosystem models and particularly for past climates and global biogeochemical cycles".

The complexity issue is certainly an important scientific question, and hopefully one that EcoGEnIE can make a contribution too. However, we feel that this is beyond the scope of the paper, as is already noted in the Discussion "The aim of this paper is to provide a detailed description of the new ecological component. It is clear from Figure 19 that the switch from the parameterised biological pump to the explicit ecological model has led to a deterioration in the overall ability of cGENIE to reproduce the global distributions of important biogeochemical tracers. This is an acceptable outcome, as our goal here is simply to provide a full description of the new model. Given that the original model was calibrated to the observations in question (Ridgwell et al., 2007a), that process will need to be repeated for the new model before any sort of objective comparison can be made".

**possible problems related to light attenuation**

This is addressed under the specific comments. Specifically, the reviewer was not correct that C:Chl will not be relevant in the model, because they vary horizontally.

**export production (no prognostic variable for POC)**

The POC equations were addressed in response to Erik Buitenhuis' comments, with our response being to change the equations. We also comment instant export term in the introduction "In the case of the nutrient-limitation models, the lack of an explicit biomass term results in export fluxes changing instantaneously in response to changing limiting factors. In the real world, by contrast, sufficient biomass must first exist, such as in a bloom condition, in order to achieve maximal export. This has 85 consequences for the how the seasonality of organic matter export is represented". To examine this further would require additional model development experiments.

**the neglect of physical transport of the ecosystem variables**

This is also addressed in the Discussion: "In the initial implementation of ECOGEM described and evaluated here, the explicit plankton community is held entirely within the ECOGEM module and is not subject to physical transport (e.g. advection and diffusion) by

the ocean circulation model (although dissolved tracers such as nutrients still are). As a first approximation, this approach appears to be acceptable, as long as the rate of transport between the very large grid cells in cGENIE is slow in relation to the net growth rates of the plankton community. On-line advection of ecosystem state variables will be implemented and its consequences explored in a future version of EcoGENIE. ”

I hope you will agree that, for the scope of the paper, we have already sufficiently addressed the main concerns.

With regard to the code availability section, I have written a tutorial for the model, which I will upload. Please let me know if this is suitable (possibly in slightly modified form).

Best regards,

Ben Ward